# Rationalizing the light-induced phase separation of mixed halide organic–inorganic perovskites

Sergiu Draguta[1], Onise Sharia[2], Seog Joon Yoon[1,3], Michael C. Brennan[1], Yurii V. Morozov[1], Joseph S. Manser[2,3], Prashant V. Kamat[1,2,3], William F. Schneider[1,2] & Masaru Kuno[1]

Mixed halide hybrid perovskites, $CH_3NH_3Pb(I_{1-x}Br_x)_3$, represent good candidates for low-cost, high efficiency photovoltaic, and light-emitting devices. Their band gaps can be tuned from 1.6 to 2.3 eV, by changing the halide anion identity. Unfortunately, mixed halide perovskites undergo phase separation under illumination. This leads to iodide- and bromide-rich domains along with corresponding changes to the material's optical/electrical response. Here, using combined spectroscopic measurements and theoretical modeling, we quantitatively rationalize all microscopic processes that occur during phase separation. Our model suggests that the driving force behind phase separation is the bandgap reduction of iodide-rich phases. It additionally explains observed non-linear intensity dependencies, as well as self-limited growth of iodide-rich domains. Most importantly, our model reveals that mixed halide perovskites can be stabilized against phase separation by deliberately engineering carrier diffusion lengths and injected carrier densities.

[1] Department of Chemistry and Biochemistry, University of Notre Dame, Notre Dame, IN 46556, USA. [2] Department of Chemical and Biomolecular Engineering, University of Notre Dame, Notre Dame, IN 46556, USA. [3] Notre Dame Radiation Laboratory, University of Notre Dame, Notre Dame, IN 46556, USA. Sergiu Draguta and Onise Sharia contributed equally to this work. Correspondence and requests for materials should be addressed to W.F.S. (email: wschneider@nd.edu) or to M.K. (email: mkuno@nd.edu)

Despite dramatic efficiency ($\eta$) increases of mixed organic–inorganic lead halide perovskite solar cells from 5 to 22%[1-4], $\eta$-values still lie below the Shockley limit[5, 6]. Ongoing efforts have therefore targeted bandgap tuning as a means of improving photovoltaic performance by better capturing the full solar spectrum[7]. Mixed halide perovskites (e.g., MAPb($I_{1-x}Br_x)_3$, where $0 < x < 1$ and MA is the methylammonium cation ($CH_3NH_3^+$)) represent an appealing approach to realizing efficient bandgap tuning. Corresponding bandgaps are linear functions of mixing fraction $x$ and span the range 1.57–2.27 eV[8]. Unfortunately, MAPb($I_{1-x}Br_x)_3$ exhibits unwanted instabilities under visible light irradiation, leading to I and Br segregation into separate iodide- and bromide-rich domains. While general aspects of photoinduced MAPb($I_{1-x}Br_x)_3$ phase separation have been established[9-16], unresolved issues remain. They include an explanation of dramatic spectral differences in the emission and absorption of phase-separated perovskites, the excitation intensity ($I_{exc}$)-dependent growth of the iodide-rich domain emission, and non-linear intensity dependencies of the phase separation rate constant. Establishing a deeper understanding of light-induced MAPb($I_{1-x}Br_x)_3$ phase separation is thus key to engineering and controlling the stability of mixed halide perovskites for photovoltaic as well as other optoelectronic applications[17-19].

Here we rationalize the bulk of experimental observations regarding the light-induced phase segregation of MAPb($I_{1-x}Br_x)_3$. From the first principles density functional theory (DFT) calculations, we explain the driving forces leading to phase segregation. We then estimate the initial size of phase-separated domains and develop an analytical model that accounts for observed non-linear intensity-dependencies in kinetic rate constants. Most importantly, we suggest how light-induced phase segregation can be mitigated in MAPb($I_{1-x}Br_x)_3$ thin films.

## Results

**Spectroscopic observations**. We first introduce spectroscopic observations that characterize the prolonged illumination of MAPb($I_{1-x}Br_x)_3$ thin films. In particular, Fig. 1a shows the evolution of a MAPb($I_{0.5}Br_{0.5})_3$ thin film emission spectrum under continuous optical irradiation ($\lambda_{exc} = 405$ nm, $I_{exc} = 20$ mW cm$^{-2}$). Evident is a decrease in the native MAPb($I_{0.5}Br_{0.5})_3$ emission at $\lambda_{mix} = 652$ nm accompanied by a corresponding rise of emission features at 725 and 527 nm. The former (latter) is associated with MAPbI$_3$-like (MAPbBr$_3$-like) photoluminescence. In what follows, the text therefore refers to the *red* (*green*) spectral feature as emission from iodide-rich (bromide-rich) domains. Furthermore, the emission growth kinetics of iodide- and bromide-rich domains are identical, as shown in Supplementary Fig. 1. Analogous behavior has been seen in films with other $x$-values (Supplementary Fig. 2).

These dramatic changes in emission are not evident in absorption. To illustrate, Fig. 1b shows the response of the MAPb($I_{0.5}Br_{0.5})_3$ spectral absorption to irradiation. A slight decrease of the band-edge extinction is accompanied by growth of a tail at longer wavelengths.

Various attempts have been made to explain these photoinduced changes to the absorption/emission of MAPb($I_{1-x}Br_x)_3$ films[10, 11]. In brief, prior DFT calculations have suggested that photoirradiation causes lattice expansion, leading to MAPb($I_{1-x}Br_x)_3$ bandgap reduction[20]. MAPbI$_3$ and MAPbBr$_3$, however, are stable upon illumination. This suggests that a more universal explanation for the phenomenon exists. Photoinduced defect formation has also been invoked to explain subgap absorption/emission in MAPb($I_{0.4}Br_{0.6})_3$ films[11]. However, the exact nature of these metastable states has never been explained.

Today, it is generally accepted that photoinduced changes to the absorption/emission of MAPb($I_{1-x}Br_x)_3$ films originates from light-induced halide phase segregation. This has been established by McGehee and co-workers using thin film X-ray diffraction measurements carried out during MAPb($I_{1-x}Br_x)_3$ illumination[10]. These measurements reveal the formation of two types of crystalline domains having lattice parameters consistent with those from iodide- (i.e., MAPb($I_{0.8}Br_{0.2})_3$) and bromide-rich (i.e., MAPb($I_{0.3}Br_{0.7})_3$) perovskite films with estimated conversion fractions ranging from $\phi = 20$ to 23%[10]. Complementary absorption measurements suggest $\phi = 18$%[12]. Temperature-dependent emission measurements have additionally yielded phase separation activation energies in good agreement with reported halide anion migration energies for inorganic perovskites (0.25–0.39 eV)[10], corroborating the above X-ray analysis.

Evidence for phase separation also comes from recent transient differential absorption measurements [12]. Namely, under external illumination a ground state bleach maximum associated with the absorption edge of MAPb($I_{0.5}Br_{0.5})_3$ disappears and is replaced by two new bleach features, one at 530 nm and another at 720 nm. The former (latter) is associated with the absorption edge of bromide-rich (iodide-rich) perovskite films. These transient absorption results notably agree with the emission data in Fig. 1a.

Thus, partial phase separation of MAPb($I_{1-x}Br_x)_3$ thin films leads to near exclusive emission from iodide-rich domains. By contrast, corresponding absorption spectra remain nearly unchanged. Beyond this, little is known about why photoirradiation leads to MAPb($I_{1-x}Br_x)_3$ phase separation, let alone why complex phase separation kinetics are observed.

Figure 1c shows that MAPb($I_{1-x}Br_x)_3$ phase separation kinetics are $I_{exc}$-dependent. Using emission data from an illuminated MAPb($I_{0.5}Br_{0.5})_3$ thin film, Fig. 1c plots $I_{iodide}/I_{sat}$ as a function of time; $I_{iodide}$ is the integrated intensity of the iodide-rich domain emission and $I_{sat}$ is an empirical saturation intensity. It is apparent that $I_{iodide}$ grows exponentially. Subsequent fitting of the data to $\frac{I_{iodide}}{I_{sat}} = \left(1 - e^{-k_{forward,em}t}\right)$ yields an apparent, emission-based, first order rate constant for phase separation ($k_{forward,em}$). In addition, Fig. 1c shows that below an excitation intensity threshold of $I_{exc} = 40$ μW cm$^{-2}$ no phase separation occurs.

Figure 1c thus makes three key points. First, $k_{forward,em}$ is excitation intensity-dependent. Namely, the rate of phase segregation increases with increasing $I_{exc}$, as seen through a correspondingly faster exponential growth of $I_{iodide}/I_{sat}$. Corresponding phase separation rate constants, shown in Fig. 1d, range from $k_{forward,em} = 0.092$–$0.823$ s$^{-1}$ for $I_{exc} = 0.26$–$58.2$ mW cm$^{-2}$. Second, $k_{forward,em}$ saturates at $k_{forward,em} = 0.707$ s$^{-1}$ when $I_{exc}$ exceeds a critical excitation intensity of 20 mW cm$^{-2}$. For comparison purposes, the associated absorption-based forward rate constant at $I_{exc} = 25$ mW cm$^{-2}$ is $k_{forward,abs} = 0.038$ s$^{-1}$ (inset, Fig. 1b). Finally, there exists an excitation intensity threshold below which no phase separation occurs.

Beyond this, there is a large asymmetry in MAPb($I_{1-x}Br_x)_3$ phase separation and recovery kinetics. In this regard, it is known that, under dark conditions, phase separation is reversible (confirmed by X-ray measurements[10]), leading to full recovery of the original MAPb($I_{1-x}Br_x)_3$ absorption and emission spectra over the course of 5 min [10-12]. The asymmetry is attributed to large activation energies for halide ion migration in the dark[10, 12, 15]. In the current study, we find an emission-based (absorption-based) recovery rate constant of $k_{reverse,em} = 6.8 \times 10^{-3}$ s$^{-1}$ ($k_{reverse,abs} = 2.3 \times 10^{-3}$ s$^{-1}$) by analyzing temporal kinetics of MAPb($I_{0.5}Br_{0.5})_3$ emission (absorption) recovery (Supplementary Fig. 3). These values agree with existing literature values on the order of $10^{-3}$ s$^{-1}$ [12] and reveal that $k_{reverse}$ is easily two orders of magnitude smaller than $k_{forward}$.

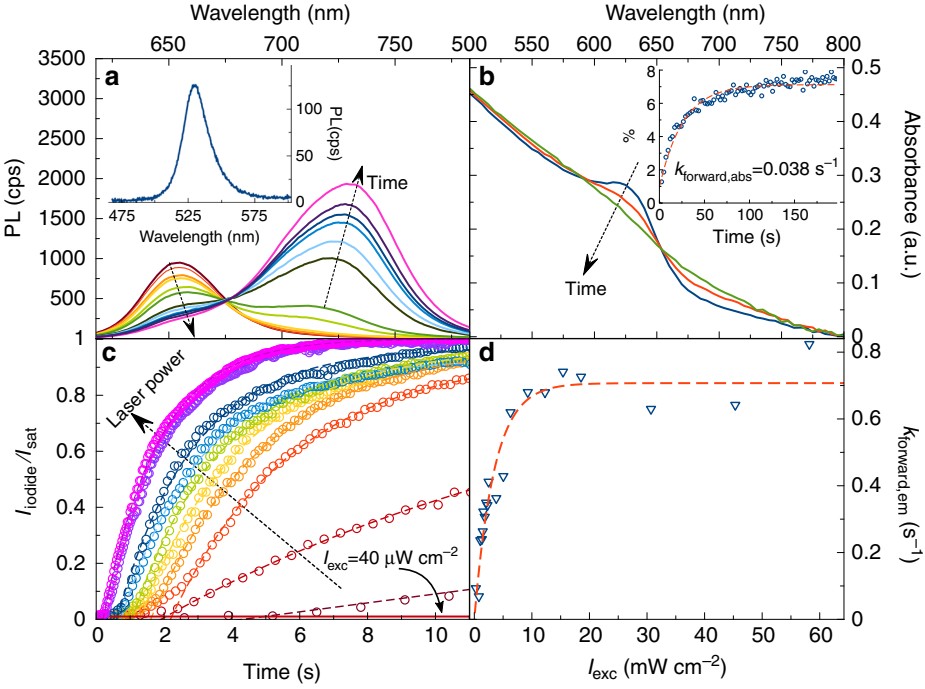

**Fig. 1** Spectroscopic observations of halide phase segregation. **a** Time evolution of MAPb(I$_{0.5}$Br$_{0.5}$)$_3$ (MA = CH$_3$NH$_3^+$) emission spectra (PL, photoluminescence) under 405 nm continuous wave excitation ($I_{exc}$ = 20 mW cm$^{-2}$). Times for selected spectra (from *red* to *purple*): 0.05, 1.41. 1.64, 1.69, 1.83, 1.93, 2.26, 2.40, 2.58, 2.68, 2.87, and 3.10 s. *Inset*: Emission spectra between 475 and 600 nm. **b** Corresponding time evolution of the MAPb (I$_{0.5}$Br$_{0.5}$)$_3$ absorption spectra under 405 nm continuous wave (CW) excitation ($I_{exc}$ = 25 mW cm$^{-2}$). Times for selected absorption spectra (from *blue* to *green*): 0, 1, 30 min. *Inset*: Absorption-based phase separation kinetics from absorption changes at 720 nm where the *dashed red line* represents an exponential fit to the data. **c** $I_{iodide}/I_{sat}$ ($I_{iodide}$ – iodide-rich emission intensity, $I_{sat}$ – saturation emission intensity) under different $I_{exc}$ ($I_{exc}$ – excitation intensity). *Dashed lines* are fits using Eq. (4). The *bottom solid red line* shows $I_{iodide}/I_{sat}$ when $I_{exc}$ = 40 µW cm$^{-2}$. Excitation intensities for selected curves (from *red* to *purple*): 0.27, 0.79, 1.05, 1.46, 1.56, 2.27, 3.81, 5.04, 18.56, 57.28 mW cm$^{-2}$. **d** $I_{exc}$-dependent emission-based first order rate constant for phase separation. The *dashed red line* represents a fit to the data using Eq. (5)

A complete microscopic model that accounts for all these observations, specifically, the absorption/emission spectral asymmetry, the $I_{exc}$-dependent forward rate constant, its saturation above a critical $I_{exc}$-value, and the existence of a threshold intensity below which no phase separation occur, remains elusive. Towards addressing these open questions, Ginsberg and co-workers have recently proposed a microscopic model that explains some of these observations[15]. In it, photogenerated polarons localize in iodide-rich regions of MAPb(I$_{1-x}$Br$_x$)$_3$ thin films, where pre-existing iodide-rich sites originate from stochastic composition fluctuations. Localized polarons induce local lattice strain, which, in turn, promotes further iodide migration to their immediate vicinity. This leads to iodide-rich domain growth. Self-limited growth arises from the finite size of the polaron deformation region and yields 8–10 nm iodide-rich domains. Subsequent emission from phase-separated films predominantly occurs from iodide-rich regions of the film due to their smaller bandgap relative to MAPb(I$_{1-x}$Br$_x$)$_3$.

Despite successes of the polaron model in rationalizing phase segregation and self-limited domain size growth, many ancillary observations, including spectral asymmetries in absorption/ emission response, as well as non-linear $I_{exc}$-dependent kinetics, have not been addressed. Consequently, a more complete accounting of light-induced MAPb(I$_{1-x}$Br$_x$)$_3$ phase segregation is needed to identify conditions that can increase the stability of MAPb(I$_{1-x}$Br$_x$)$_3$ films under illumination.

**Phase segregation model.** In what follows, we describe a conceptual framework that rationalizes virtually all current experimental observations regarding MAPb(I$_{1-x}$Br$_x$)$_3$ phase separation.

The framework consists of a kinetic model that describes events following MAPb(I$_{1-x}$Br$_x$)$_3$ photoexcitation. Implicit assumptions of the kinetic model are supported by first principles DFT calculations. The kinetic model is subsequently accompanied by analytical (probabilistic) estimates to describe the emission from phase-separated MAPb(I$_{1-x}$Br$_x$)$_3$ films.

Figure 2a illustrates microscopic processes that underpin the kinetic model. First, light absorption creates an electron–hole pair. Along Path 1, these carriers recombine to produce native MAPb(I$_{1-x}$Br$_x$)$_3$ emission ($\lambda_{mix}$ = 652 nm, $x$ = 0.5). Along Path 2, photoexcitation provides the driving force to induce local halide anion rearrangement and microscopic MAPb(I$_{1-x}$Br$_x$)$_3$ phase segregation into iodide- and bromide-rich domains. Although the exact final composition of these iodide-rich domains is not known, Hoke et al. have suggested the existence of a stable iodide-rich phase, following prolonged illumination, with a nominal composition of $x$~0.2[10]. Subsequent emission occurs from these iodide-rich domains (Path 2a, $\lambda_{iodide}$) due to their favorable band offsets. In tandem, the phase separation reverses on significantly slower timescales, along Path 2b[12]. Finally, Path 3 considers the possibility of MAPb(I$_{1-x}$Br$_x$)$_3$ photogenerated carriers diffusing to existing iodide-rich domains on length scales comparable to electron/hole diffusion lengths (ca. $l_{e/h}$ = 100 nm)[21]. Consequently, emission can occur from iodide-rich regions of phase-segregated films even when photoexcitation events themselves do not initiate phase segregation. Supplementary Note 1 provides a complete mathematical accounting of the model.

DFT calculations detailed at length in Supplementary Note 2 (see Supplementary Fig. 4) support major assumptions underlying the kinetic model. Specifically, calculations in four formula

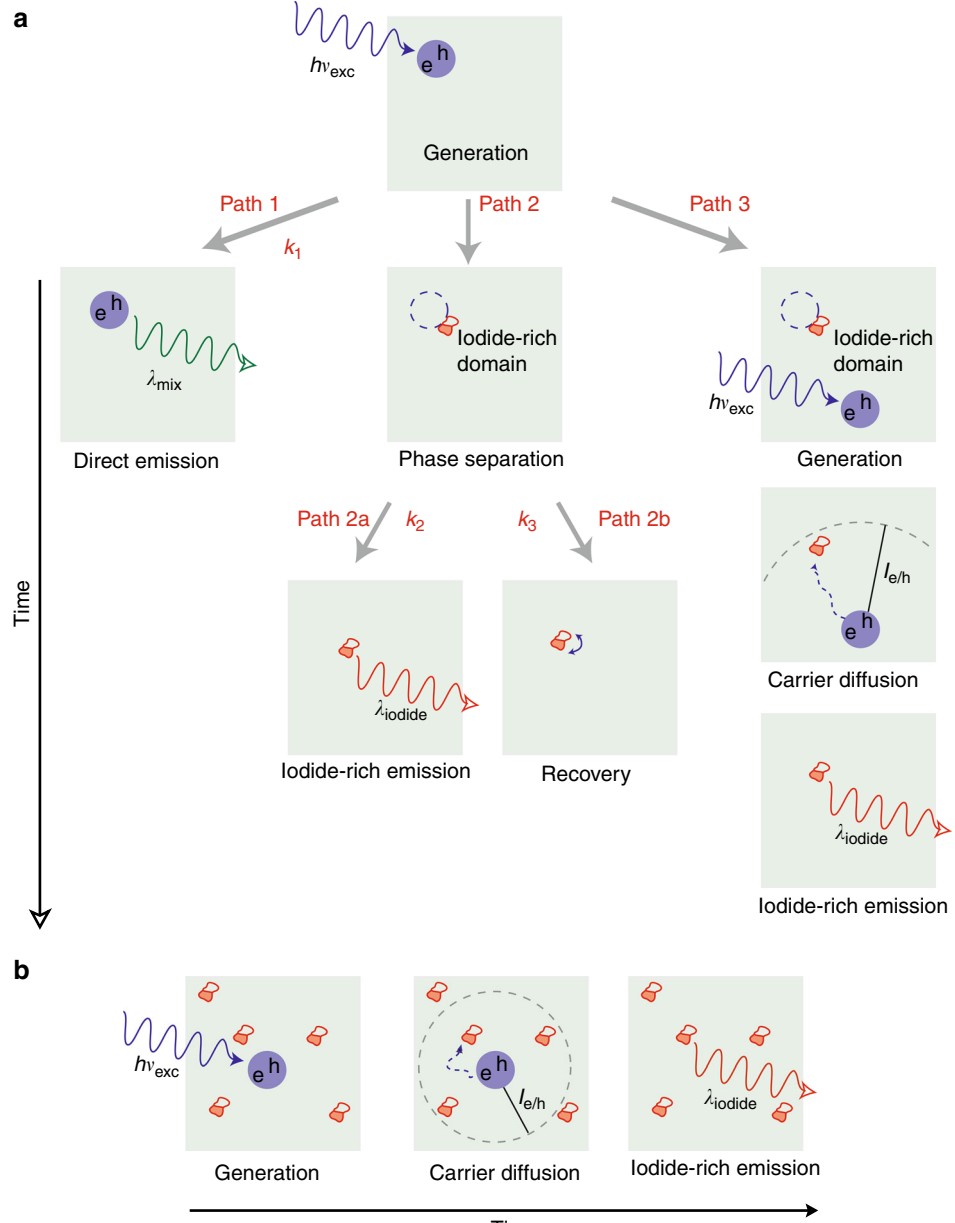

**Fig. 2** Schematic of relevant kinetic processes during halide phase segregation. **a** low and **b** high excitation intensity light-induced phase separation of MAPb(I$_{1-x}$Br$_x$)$_3$ (MA = CH$_3$NH$_3^+$). Microscopic rate constants ($k_{1-3}$) associated with the underlying kinetic model have been provided. *Dark circles* denote photogenerated electron–hole pairs. *Empty circles* denote electron–hole pairs which have induced phase separation. *Filled red* and *white* regions represent phase separated iodide-rich and corresponding bromide-rich domains

unit (FU) supercells show that the energy ($\Delta E_{GS}(x)$) to form mixed MAPb(I$_{1-x}$Br$_x$)$_3$ from MAPbI$_3$ and MAPbBr$_3$ is small and slightly positive (on the order of 0.01 eV FU$^{-1}$) up to $x = 1/3$ and modestly negative at larger $x$ (Fig. 3a, *blue circles*). In this limit of nearly zero formation energies, mixing entropy controls phase stability. At ambient temperature, the ideal entropy of halide mixing substantially exceeds the mixing energy, so that, as illustrated in Fig. 3a (*green squares*), the mixed phase minimizes free energy ($\Delta F_{GS}(x)$) over a substantial composition range. Brivio et al.[9] report DFT-computed mixing energies of similar magnitude and nearly ideal mixing entropies over the entire composition range. Both results therefore imply that over the composition range considered, entropic considerations lead to MAPb(I$_{1-x}$Br$_x$)$_3$ formation.

Next, we use DFT calculations to estimate the ground state valence band position of MAPb(I$_{0.5}$Br$_{0.5}$)$_3$ relative to those of MAPbI$_3$ and MAPbBr$_3$. This comparison is shown in Supplementary Fig. 5, which reveals lower (higher) valence band edges within iodide-rich (bromide-rich) domains. Further, these calculations and available experiment[22, 23] indicate that mixed and iodide-rich conduction band edges are nearly isoenergetic. Altogether, these results imply that photogenerated holes will preferentially localize in iodide-rich regions of the film and will ultimately contribute to red-shifted, MAPbI$_3$-like emission (Path 2, Fig. 2a).

This valence edge difference provides an energy driving force for demixing that counterbalances the above entropic preference for halide mixing. We can estimate this trade-off by expressing the phase separation energy between a photoexcited hole in MAPb(I$_{1-x}$Br$_x$)$_3$ and a hole localized in an iodide-rich phase. Because of their near isoenergetic conduction band edges, experimentally-observed MAPb(I$_{1-x}$Br$_x$)$_3$/MAPbI$_3$ band gap

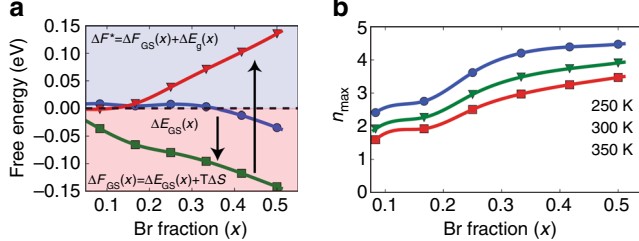

**Fig. 3** Density functional theory modeling of halide phase segregation. **a** Estimated free energy of formation as a function of bromide fraction $x$. The *blue line/circles* are generalized gradient approximation (GGA)-computed 0 K ground state formation energies, including spin-orbit coupling. The *green line/squares* are free energies of mixing at 300 K, assuming ideal mixing on the halide sublattice. The *red line/inverted triangles* represent the free energy difference after single photon absorption. All energies are reported per two formula units. *Blue* and *pink* shaded areas indicate positive (phase separation favored) and negative (phase separation disfavored) free energy regions. **b** Approximate size of separated domains, $n_{max}$, following single photon absorption

differences[8] $[\Delta E_g(x)]$, which increase with $x$, capture the demixing driving force. Consequently, the free energy change associated with the transformation of $n$ formula units of photoexcited MAPb$(I_{1-x}Br_x)_3$ into pure bromide and pure photoexcited iodide $(\Delta F^*)$ can be written as

$$\Delta F^*(x, n) = n\left(F_{mix}^*(x) - xF_{Br} - F_I^*\right) = n\Delta F_{GS}(x) + \Delta E_g(x) \tag{1}$$

where $F_{mix}^*$, $F_{Br}$, and $F_I^*$ denote photoexcited MAPb$(I_{1-x}Br_x)_3$, MAPbBr$_3$, and MAPbI$_3$ free energies and asterisks denote the presence of a photogenerated hole. $\Delta F^*$ is plotted in Fig. 3a (*inverted triangles*) for $n = 2$ at $T = 300$ K. Note that separation into iodide- and bromide-enriched phases yield correspondingly smaller, yet positive, $\Delta F_{GS}$ values. In this regard, Fig. 3a shows $\Delta F^*$ to be close to zero in the composition region $x < 0.1$ with a predominantly iodide-rich composition of MAPb$(I_{0.9}Br_{0.1})_3$. Although the estimate is consistent with the results of McGehee et al.[10] who have previously found iodide-rich phases with $x \sim 0.2$ (i.e., MAPb$(I_{0.8}Br_{0.2})_3$), following prolonged photoexcitation, McGehee et al.[10], as well as Noh et al.[8] have also suggested that such iodide-enriched phases undergo a crystallographic phase transition from cubic to tetragonal. It is therefore possible that such a phase transition could stabilize an iodide-rich phase. Consequently, an exact quantitative accounting of light-induced phase separation, especially in the iodide-rich limit, would require both crystallographic forms to be considered explicitly within the current DFT and thermodynamic modeling[8, 10].

An approximate estimate for the initial domain size of phase-separated regions can be found from Eq. (1), which vanishes at $n = n_{max}$, the maximum number of formula units capable of being phase separated by a single hole:

$$n_{max}(x) = -\frac{\Delta E_g(x)}{\Delta F_{GS}(x)} \tag{2}$$

Figure 3b plots Eq. (2) and reveals that for compositions $0.083 < x < 0.5$ and for temperatures $250$ K $< T < 400$ K, $n_{max}$ ranges from 2 to 4. Exact $n_{max}$ values are sensitive to the actual composition of iodide- and bromide-rich phases and to the DFT energies. Consequently, they are difficult to predict precisely. Both Eqs. (1) and (2), however, point to the important role played by MAPb$(I_{1-x}Br_x)_3$/MAPbI$_3$ band gap differences in driving phase segregation under optical illumination.

At this point, returning to the kinetic model, we note that the existence of sizable MAPb$(I_{1-x}Br_x)_3$ electron/hole diffusion lengths $(l_{e/h} = 100$ nm)[21] means that photogenerated carriers created elsewhere in the film can readily access iodide-rich domains (Path 3, Fig. 2a). Consequently, radiative recombination in irradiated MAPb$(I_{1-x}Br_x)_3$ films will be dominated by MAPbI$_3$-like emission given (a) the existence of an intrinsic driving force for carriers to localize in iodide-rich domains, (b) the propensity of photoexcitation to induce MAPb$(I_{1-x}Br_x)_3$ demixing, and (c) sizable $l_{e/h}$ values.

One of the more important outcomes of the kinetic model is an expression for the phase-segregated iodide-rich fraction, $\phi_I(t)$, in the film

$$\phi_I(t) \cong \frac{1}{2}\left[1 - e^{-k_{forward,abs}t}\right] \tag{3}$$

where $\phi_I(t) \approx k_{forward,abs}t$ at early times. The link between $\phi_I(t)$ and the absorption-based forward rate constant for phase separation ($k_{forward,abs}$) is made because the absorption measurement directly reports on the phase separation. Eq. (3) is derived in Supplementary Note 1. Using $k_{forward,abs} = 0.038$ s$^{-1}$ we find that phase separated fractions nominally start from a few percent and eventually grow to values on the order of 10%. The kinetic model, backed by DFT, thus rationalizes apparent emission/absorption asymmetries in Fig. 1a, b wherein partial phase segregation results in near exclusive iodide-rich emission despite little to no change of the absorption.

We now consider the origin of the $I_{exc}$-dependent phase separation kinetics seen in Fig. 1c, d. This entails invoking a probabilistic consideration of the emission from iodide-rich domains with a focus on contributions from carrier diffusion (Path 3, Fig. 2a). In this regard, iodide-rich emission originating from diffusion becomes especially important at high excitation intensities, because carrier generation occurs in the presence of multiple segregated iodide-rich domains. A sizable probability therefore exists for carriers to find such a domain within a geometric volume defined by $l_{e/h}$. This probability increases with increasing excitation intensity due to the slow timescale for phase separation recovery[12] and naturally introduces an $I_{exc}$-dependence to $I_{iodide}$. Figure 2b outlines the scenario.

Conceptually, we assume that iodide-rich domains are Poisson distributed within a given carrier diffusion volume, $V_D = \frac{4}{3}\pi l_{e/h}^3 = 4.18 \times 10^6$ nm$^3$. We additionally assume that diffusing electrons and holes recombine with unit probability from an iodide-rich domain if one exists within $V_D$. From this, we obtain the following expression for the emission intensity of iodide-rich domains

$$\frac{I_{iodide}(t)}{I_{sat}} = \left[1 - e^{-\frac{V_D}{\overline{V}}\phi_I(t)}\right] \approx \left[1 - e^{-k_{forward,em}t}\right] \tag{4}$$

$I_{sat}$ is an empirical emission saturation intensity first seen in Fig. 1c, $\overline{V}$ is the average volume of individual phase-separated domains and $k_{forward,em} \approx \frac{V_D}{\overline{V}}k_{forward,abs}$ at short times. Equation (4) is derived in Supplementary Note 1. Of specific note, Eq. (4) immediately predicts that $I_{iodide}(t)$ grows exponentially. Furthermore, because $l_{e/h}$ is much greater than $l$ (i.e., $V_D$ is much greater than $\overline{V}$), $I_{iodide}$ saturates even for small values of $\phi_I(t)$. The qualitative behavior seen in Fig. 1c is therefore immediately rationalized.

Next, we compare the phase-separated domain sizes, estimated using the relationship between $\phi_I(t)$ and $k_{forward,em}$ (Eq. (4)), to experiment. In particular, from Eqs. (3) and (4) it can be shown that $l \cong \left(\frac{k_{forward,abs}}{k_{forward,em}}\right)^{1/3} l_{e/h}$. Using $l_{e/h} = 100$ nm, $k_{forward,abs} =$

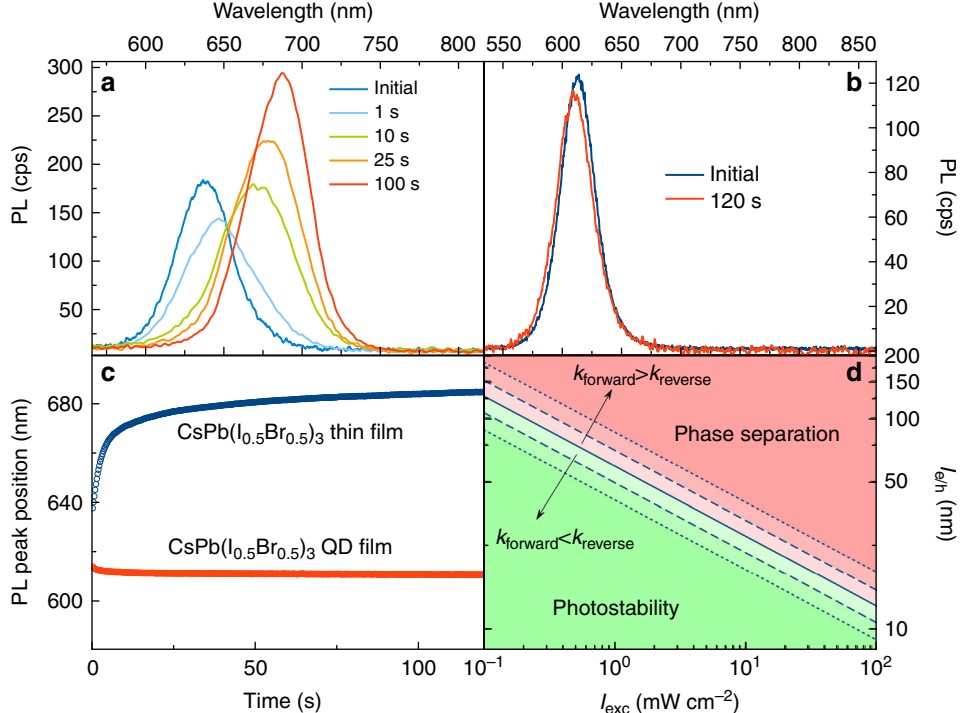

**Fig. 4** Halide phase segregation versus photostability. Time evolution of $CsPb(I_{0.5}Br_{0.5})_3$ **a** thin film and **b** nanocrystal-based film emission spectra (PL, photoluminescence) under 405 nm continuous wave (CW) excitation ($I_{exc} = 60$ mW cm$^{-2}$). **c** $CsPb(I_{0.5}Br_{0.5})_3$ thin film and nanocrystal-based film emission peak position during illumination. **d** Photostability phase diagram of $MAPb(I_{1-x}Br_x)_3$ (MA = $CH_3NH_3^+$) obtained from Eq. (5)

0.038 s$^{-1}$ (Fig. 1b) and $k_{forward,em} = 0.7$ s$^{-1}$ (Fig. 1a), we estimate a long-timescale domain size of $l = 37$ nm. This order of magnitude length scale agrees well with the experimental observations of McGehee[10] and Ginsberg[15]. However, it is larger than initial sizes linked to DFT-estimated $n_{max}$ values (below 1 nm). The discrepancy is readily explained by invoking domain growth under prolonged illumination. Qualitatively, domain growth arises because phase separation recovery rates depend on specific surface area and hence increase with decreasing domain size. In support of this, ref.[12] reports a decrease of the recovery rate with increasing sample illumination time.

We now explain the non-linear $I_{exc}$-dependence of $k_{forward,em}$ in Fig. 1d. Empirically, this behavior suggests that $k_{forward,em}$ is a non-linear function of photogenerated carrier density ($\rho$). On examining the link between $k_{forward,em}$ and $\phi_I(t)$ in Eq. (4) and invoking Eq. (3) it is apparent that $k_{forward,em}$ takes the functional form $k_{forward,em} = k_{sat}[1 - e^{-\beta I_{exc}}]$, where $k_{sat}$ is an empirical saturation rate constant and $\beta$ is a fitting parameter. Physically, this implies that the probability of creating a phase-separated domain within a geometric volume, $V_D$, remains the same irrespective of the number of photogenerated carriers already in that volume. Consequently, $k_{forward,em}$ can be derived independently by assuming that it is proportional to the probability that at least one carrier exists within $V_D$. As described in Supplementary Note 3, the result is

$$k_{forward,em} = k_{sat}[1 - e^{-\beta I_{exc}}] = k_{sat}[1 - e^{-V_D\rho}]. \quad (5)$$

Subsequent fitting of the data in Fig. 1d with Eq. (5) yields good agreement between model (*dashed red line*) and experiment.

**Phase segregation versus photostability.** Beyond qualitatively rationalizing the data, the most significant aspect of Eq. (5) is that it is predictive. This is because, Eq. (5) suggests that $k_{forward,em}$ can

be tuned by varying either $I_{exc}$ or $MAPb(I_{1-x}Br_x)_3$ carrier diffusion lengths. To illustrate, Eq. (5) can be used to estimate an intensity threshold below which no phase separation occurs. By finding $I_{exc}$, where $k_{forward,em}$ is approximately an order of magnitude smaller than $k_{reverse,em}$, a situation is established where phase separation becomes kinetically unfavorable and where any newly formed iodide- and bromide-rich domains immediately recover to $MAPb(I_{1-x}Br_x)_3$. As shown in Supplementary Note 3, this occurs when $I_{exc} = 20\,\mu$W cm$^{-2}$. Remarkably, this value is in excellent agreement with the experimentally-determined threshold value of $I_{exc} = 40\,\mu$W cm$^{-2}$ in Fig. 1c.

Alternatively, Eq. (5) shows that lowering $l_{e/h}$ (i.e., $V_D$) can yield $k_{forward,em}$ values comparable to $k_{reverse,em}$. Under these conditions, phase separation in $MAPb(I_{1-x}Br_x)_3$ films again becomes kinetically unfavorable. To directly test this latter prediction, we have fabricated $CsPb(I_{0.5}Br_{0.5})_3$ thin films along with their complementary nanocrystal-based films. Electron/hole diffusion lengths are shorter in nanocrystal-based films than in their thin film counterparts because grain boundaries between nanocrystals suppress charge transport[24]. Cesium lead halide perovskites have additionally been chosen because of the better size and composition control of corresponding nanocrystals[25]. Supplementary Note 4 details the synthesis of $CsPbBr_3$ nanocrystals and thin films (see also Supplementary Figs. 6 and 7, Supplementary Table 1).

Figure 4a shows that $CsPb(I_{0.5}Br_{0.5})_3$ thin films exhibit similar PL spectral redshifts as their $MAPb(I_{1-x}Br_x)_3$ counterparts under illumination ($\lambda_{exc} = 405$ nm, $I_{exc} = 60$ mW cm$^{-2}$). An initial emission peak at $\lambda_{mix} = 637$ nm decreases in intensity and is accompanied by prompt growth of a second feature at 687 nm. As before, this redshift arises due to halide phase separation[7].

By contrast, Fig. 4b reveals that no redshift occurs in $CsPb(I_{0.5}Br_{0.5})_3$ nanocrystal-based films under the same excitation conditions. If anything, a small 3 nm blueshift is observed, which does not arise from phase separation, as pure $CsPbBr_3$ thin films

respond identically under optical excitation. The blueshift thus likely relates to sample instabilities under ambient conditions. Figure 4c compares the emission peak position of a CsPb $(I_{0.5}Br_{0.5})_3$ thin film and a CsPb$(I_{0.5}Br_{0.5})_3$ nanocrystal-based film under illumination.

As an added control to verify that phase separation indeed arises from suppressed charge diffusion lengths and not from the limited nanocrystal size in nanocrystal-based films, we show in Supplementary Fig. 8 that at high excitation intensities ($I_{exc} = 500$ W cm$^{-2}$) phase separation in nanocrystal-based films can be turned on. We have additionally annealed nanocrystal-based films to sinter component nanoparticles together and have shown that sintered films behave in the same manner as their thin film MAPb$(I_{0.5}Br_{0.5})_3$ counterparts. At low excitation intensities (i.e., $I_{exc} = 2.5$ W cm$^{-2}$) no phase separation occurs. At high excitation intensities (i.e., $I_{exc} = 55$ W cm$^{-2}$), clear phase separation-induced redshifting of the emission becomes evident (Supplementary Fig. 9).

## Discussion

We conclude that phase separation in mixed halide perovskites can be controlled by deliberately reducing either $I_{exc}$ or carrier diffusion lengths. Sensitivity to either parameter is predicted by Eq. (5) such that depending on application, a balance between the two can be found in order to improve the stability of mixed halide perovskites under illumination. Towards this, Fig. 4d provides a MAPb$(I_{1-x}Br_x)_3$ phase segregation stability diagram constructed from Eq. (5). Assuming that stability occurs when $k_{forward}$ is less than $k_{reverse}$, the condition $k_{forward} = k_{reverse}$ sets the boundary between phase segregation and photostability. Employing $k_{reverse} = 6.8 \times 10^{-3}$ s$^{-1}$ (Supplementary Fig. 3) and $k_{sat} = 0.7$ s$^{-1}$ (Fig. 1d) yields complementary $l_{e/h}$ and $I_{exc}$ pairs which demark the MAPb $(I_{1-x}Br_x)_3$ optical stability boundary (*solid line*, Fig. 4d). Green regions show parts of the $l_{e/h}/I_{exc}$ phase diagram photostable under illumination. Associated red regions reveal conditions where phase segregation occurs.

We find that for MAPb$(I_{1-x}Br_x)_3$ thin films under one sun illumination ($I_{exc} = 100$ mW cm$^{-2}$) $l_{e/h}$ should be smaller than 13 nm to suppress phase segregation. This insight could therefore aid the development of mesoporous hybrid perovskite solar cells where carrier diffusion lengths are already small and where power conversion efficiencies are currently $\eta = 19\%$[26]. In addition, while reducing $l_{e/h}$ in corresponding planar architectures may not be optimal for improving $\eta$, the choice of cation (e.g., cesium, formamidinium) also affects $k_{reverse}$. Hence, this represents an alternative way by which to enhance the photostability of mixed halide perovskite films in planar devices[7, 27].

In summary, we have established quantitative insights into the light-induced phase separation of MAPb$(I_{1-x}Br_x)_3$ thin films. Bandgap reduction of iodide-rich domains is found to be the driving force that overcomes unfavorable formation energies to induce iodide and bromide segregation. A DFT-based thermodynamic model shows that entropy dominates formation free energies and provides estimates of the initial phase-separated domain size and composition. These predictions are sensitive to precise structural and entropic models, and quantitative predictions, especially in the iodide-rich limit, would demand a much broader survey of the perovskite crystallographic structure and composition spaces. In particular, it is possible that a crystallographic phase transition additionally stabilizes an iodide-rich phase. Initial phase-separated domains undergo self-limited growth to final sizes on the order of 30 nm. Despite partial phase segregation, the emission properties of MAPb$(I_{1-x}Br_x)_3$ films are near exclusively dictated by nucleated iodide-rich domains. This stems from favorable energetics for hole localization and from the existence of large electron/hole diffusions

lengths. Further modeling reveals that MAPb$(I_{1-x}Br_x)_3$ films can be stabilized against phase segregation by reducing carrier diffusion lengths and/or excitation intensities with a phase stability diagram indicating complementary $I_{exc}$ and $l_{e/h}$ pairs yielding photostability. Insights from this study thus set the basis for future experiments to engineer and ultimately control the stability of mixed halide perovskites for photovoltaic as well as other optoelectronic applications.

## Methods

**Density functional theory**. Plane-wave, supercell calculations were performed in the Vienna Ab initio Simulation Package (VASP)[28] within the generalized gradient approximation (GGA)[29] and the projector-augmented wave (PAW)[30] treatment of core states. Valence electrons were expanded to a 500 eV cut-off. Mixed halide structures were computed within orthorhombic supercells of approximate size $9.3 \times 12.7 \times 8.6$ Å with the first Brillouin zone sampled using a $4 \times 4 \times 4$ **k**-point mesh. Atomic positions and lattice parameters were relaxed simultaneously using the conjugate gradient method. Spin-orbit coupling and non-collinear magnetism were included in the total energy calculations. Further details can be found in the Supplementary Note 2.

**Sample preparation**. All sample processing was conducted in a N$_2$ filled glove box. A 2 ml solution consisting of: CH$_3$NH$_3$Br (Dyesol) (0.15 M, 34 mg), CH$_3$NH$_3$I (Dyesol) (0.15 M, 48 mg), PbBr$_2$ (98%, Alfa Aesar) (0.15 M, 110 mg), and PbI$_2$ (99.9985%, Alfa Aesar) (0.15 M, 138 mg) in N,N-dimethylformamide (DMF, 99.8%, Sigma-Aldrich) was used as a precursor solution. A 0.2 μm PTFE filter was used to remove any particulates. Precursors were spincoated onto cleaned cover glass (Vetrini Coprioggetto) at 5500 rpm for 30 s with a 1000 rpm s$^{-1}$ ramp rate. Polymethylmethacrylate (PMMA, Alfa Aesar, MW = 450–550 K) was subsequently deposited over the film by spincoating a 10 mg/ml solution in anhydrous chlorobenzene (2000 rpm for 30 s, 1000 rpm s$^{-1}$ ramp rate). Resulting films were annealed on a hot plate at 100 °C for 5 min

**Emission measurements**. Emission spectra were collected with an electron multiplying charged coupled device (EMCCD) camera (Andor) connected to a spectrometer (Acton). A continuous wave 405 nm laser (Coherent, Obis) was used as the excitation source. For power-dependent emission intensity measurements, spectra were collected with variable EMCCD integration times, ranging from 0.1–0.4 s. All emission spectra were fit to two Gaussians (one for $\lambda_{mix}$ and the other for $\lambda_{iodide}$) to estimate the integrated emission intensity of iodide-rich domains ($I_{iodide}$). Integrated emission intensities of bromide-rich domains were obtained using both a longpass (425 nm, Chroma) and a shortpass (532 nm, Semrock) filter coupled to an avalanche photodiode (Perkin Elmer, SPCM-AQR-14).

For emission-based phase segregation recovery measurements, samples were first irradiated with a CW laser ($\lambda_{exc} = 520$ nm, Coherent Obis) with an excitation intensity of $I_{exc} = 20$ mW cm$^{-2}$ for 1 min to induce phase segregation. Following this, the laser was turned off. It was subsequently turned on periodically for 50 ms at a significantly lower intensity ($I_{exc} = 50$ μW cm$^{-2}$) to acquire PL spectra during recovery under dark conditions. Emission spectra were acquired using a fiber-based spectrometer (Ocean Optics).

**Absorption measurements**. Complementary absorption measurements during phase separation were conducted by illuminating films with a 405 nm CW laser ($I_{exc} = 25$ mW cm$^{-2}$) and periodically acquiring absorption spectra over the course of 30 min using a commercial spectrometer (Varian).

**Data availability**. The data that support the findings of this study are available from the corresponding authors (M.K. or W.S.) upon request.

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

## Acknowledgements

This work was supported by the Division of Materials Sciences and Engineering, Office of Basic Energy Sciences, U.S. Department of Energy under award DE-SC0014334. P.K. and S.Y. acknowledge support from the Division of Chemical Sciences, Geosciences, and Biosciences under award DE-FC02-04ER15533, Office of Basic Energy Sciences, U.S. Department of Energy. We also acknowledge ND Energy at the University of Notre Dame for seed funding. J.M. acknowledges support of King Abdullah University of Science and Technology (KAUST) through the Award OCRF-2014-CRG3-2268. We also thank the ND Colleges of Science and Engineering for financial support. This is contribution number NDRL No. 5138 from the Notre Dame Radiation Laboratory.

## Author contributions

O.S. and W.F.S. developed the theoretical model; S.D., S.Y., and Y.V.M. performed the experiments; S.Y. prepared MAPb(Br$_x$I$_{1−x}$)$_3$ thin film samples, while M.B. prepared CsPb (I$_{1−x}$Br$_x$)$_3$ nanocrystals/thin films. O.S., S.D., J.M., M.K., and W.F.S. analyzed the data, O.S., S.D., P.K., W.F.S., and M.K. co-wrote the paper.
