## [Peer Review File · Nature Communications]

Reviewers' comments:

Reviewer #1 (Remarks to the Author):

The manuscript entitled "Rationalizing the light-induced phase separation of mixed halide organic-inorganic perovskites" by Draguta et al studies the effect of light exposure on the photoluminescence and structural properties of mixed iodide-bromide perovskites, $\text{MAPb}[\text{I}(\text{x})\text{Br}(1-\text{x})]_3$ using a combination of experiment and ab-initio calculations. Since the emergence of various combinations of mixed halide/mixed cation perovskites as the lead contenders for high performance PV, their study is timely and of importance to the field.

After revising this paper I believe that it does warrant publication, however I am not convinced that his manuscript is a good fit for Nature Communications.

Starting with the presentation, I do think that the structure of the manuscript would benefit from some reshaping; currently I do not find the presentation accessible to a general readership as Nature Communications publications require. In particular I think the description of the fitting models used for rationalizing the phase-segregation process are not very clear and are quite difficult to follow.

Aside from the presentation critiques, there are a few technical questions mainly concerned with the atomistic modelling side which I think must be addressed before the article could be considered for publication.

a) The authors specify that the formation energies of the mixed halide perovskites are calculated using DFT (Figure S3). How are the mixed phases constructed for each concentration? From the computational details authors mention that 'supercells' of $9.3 \times 12.7 \times 8.6$ angstrom were used in calculations. However, this is in fact not a supercell, but actually (roughly) the size of the unit cell of MAPb in the orthorhombic phase (as measured by Baikie et al, J Mater Chem A 2013, 1, 5628-5641). It is not clear from the text if the authors use this structure to simulate the mixed compounds, but if so, this approach would bring about significant errors, because they would in fact impose through periodic boundary conditions an ordered arrangement of Br/I configurations which is not evidenced experimentally. The only two ways to ensure such errors do not occur are either to consider (real) supercells of increasing sizes and study multiple (symmetry inequivalent) configurations, or to employ the virtual crystal approximation.

b) Another problem in the electronic structure calculations of MAPb is the orientation of the MA cations - how do they decide on the orientation of the MA cation in both the separate calculations of MAPb and MAPb, and the mixed compound? The choice for orientation of the MA cation can distort significantly the structure of the PbX_6 network and can sometimes even induce large changes in the electronic structure. In particular, the band alignment shown in Figure S3, which looks to have a valence band offset of 0.1-0.2 eV is highly questionable without a comprehensive discussion and evidence on how these atomistic models were actually built, and how the choices made actually impact the computational conclusions.

c) In the section describing the computational methods the authors specify they have/have not used spin-polarized calculations. Why are the authors even considering spin-polarization? MAPb is known to a strong spin-orbit coupling effect (which requires treatment for non-collinear spins), which does reduce the band gap by 1 eV. The authors should justify their choices and/or correct their calculations in this sense.

d) In Supporting note 3: 'We estimate the maximal value of n for which the hole localization in the I-rich phase overcomes the free energy of mixing by Eq.S17' – how are the authors reaching this expression?

e) In Supporting note 3 (and through the text) authors use DFT to understand a light-induced process. There is a fundamental problem here in that DFT is a ground state theory – the type of calculations described by the authors do not at any point include excitations (of light or any kind) in the formalism. The authors make no comment on what the implications of this (quite severe approximation) will have on their results, and indeed their conclusions. Therefore, it is really unclear how the authors believe that DFT calculations can in fact provide any support to the experimental data shown throughout the paper.

Reviewer #2 (Remarks to the Author):

In this manuscript the authors advance the understanding of why there is light -induced phase segregation in mixed halide perovskite semiconductors. The subject is highly important and of interest to the readers of Nature Communications.

As the manuscript is currently written, I am not able to fully understand what the authors are trying to say. I think the manuscript can likely be modified and become suitable for publication. Below are a list of issues to address.

1. Γ (subscript Γ) is not well defined at the point where it is first used.
2. I do not think Ginsberg et al would agree with the statement "The polaron model additionally suggests that decreasing defect densities in $\text{MAPb}(\text{I}1-\text{xBrx})_3$ films should suppress phase separation. However, McGehee and coworkers have previously observed phase separation in single crystals, which exhibit smaller defect densities than their polycrystalline thin film counterparts." The polarons in Ginsberg's model are photogenerated charge carriers, not defects.
3. I do not think I understand figure 2. I am not 100% sure what a dark circle is supposed to represent. I have no idea at all what a light circle is supposed to represent. I do not know what the red circles are either.
4. Line 238 states: "This resulting value is in excellent agreement with the experimental results of McGehee and Ginsberg.¹⁵" Since McGehee and Ginsberg were not coworkers and published separate papers, both papers need to be referenced to avoid confusion.
5. Most of the explanation provided in the manuscript on why the phase separation occurs was already stated in the seminal paper by Hoke, McGehee et al. What remains to be explained is why it is thermodynamically favorable for the perovskite to be mixed in the dark, but phase separated in the light. Brivio et al's DFT calculations fail to explain this reversibility. Ginsberg's Arxiv paper provides a plausible explanation, but is extraordinarily difficult to understand and has not been accepted in a pure-review journal yet. I consider supplementary figure S3 to be the most important part of this manuscript. I certainly would put it in the main part of the manuscript. Unfortunately I do not consider the calculations that go into figure S3 to be adequately explained. It would seem to me that the increase in free energy due to the presence of excited charge carriers would depend on the density of excited charge carriers. I am not able to tell what assumptions regarding this density were made. The calculations to be explained better.
6. I don't think the take-home message of this manuscript should be that one can avoid the light - induced phase segregation by reducing the charge carrier diffusion length. For a solar cell to be efficient, it must operate at a high voltage, which implies that there is substantial quasi-Fermi level splitting, which in turn implies a high carrier density. One cannot have a good solar cell without having a high carrier density. If there is a high carrier density, the phase separation is going to occur. Improving the stability by making an inefficient solar cell is not a path that we probably want to go down.

Reviewer #3 (Remarks to the Author):

In this manuscript, the authors investigate the changes in optical properties of a mixed halide methyl ammonium lead perovskite. To explain their observations, they claim reversible phase segregation for which they find support in DFT calculations performed at differently composed

samples. Thermodynamic instability of the excited material is claimed as opposed to thermodynamic stability in the dark. It is claimed that electronic states in phase-segregated parts are selectively occupied by charge carriers diffusing in the material. Therefore, it is claimed that the materials can be stabilized by limiting the size of crystalline domains below a critical value. This is experimentally shown for a mixed halide caesium lead perovskite.

The work is highly original, dwells on solid links to recently published work, provides sufficient detail to reproduce the work and will be of great relevance to an extensive readership interested in the attractive photovoltaic properties of such materials.

Some aspects of the work, however, are not fully convincing and deserve the authors' attention before the work can be published:

1. (p.3) The authors claim the presence of an emission signal at 527 nm which is, however, not shown in Figure 1a (no data below 620 nm) but also is not present in Figure S2a which clearly expands into that range. Further, the authors argue in favour of a dominance of emission at 725 nm over all other signals later in the manuscript. To the opinion of this reviewer such inconsistency should be avoided.

2. (eq.1) Formation of pure MAPbI_3 is claimed whereas ref 10 reported formation of $\text{MAPbI}_{2.4}\text{Br}_{0.6}$ for all compositions with less than 12.4. Indeed the formation of such composition would even be suggested by the authors' report of (almost) stable emission spectra for 12.64 and 12.34 in Figure S1 b and c. Further, disappearance of a well-defined band edge (Figure 1b) without establishment of any other edge characteristic for newly formed perovskite phases would speak in favour of defect formation disturbing the well-defined perovskite band structure and leading to a distribution of superimposed apparent band edges.

3. The predictive power of the present DFT calculations is highly overestimated. The authors calculate without consideration of spin polarization. For heavy atoms like Pb this will lead to significantly wrong energy values. The small differences of less than 0.3 eV (e.g., Figure S3) discussed by the authors can easily be within the margin of error of apparent energy differences. Further (main point of criticism !) well-known structural relaxations of methyl ammonium lead iodides/bromides at room temperature are not considered in the present work. The transition from the tetragonal to the cubic phase as most stable structure was observed at 12.4. It would be very surprising if it was just coincidence that the authors observe a very similar composition as critical for the changes observed in the optical properties. The DFT calculations, however, would need to include MD simulation in order to be able to consider such structural relaxations at $T > 0\text{K}$.

4. While it would be of great technical interest to be able to stabilize mixed halide lead perovskites of a given composition, it would certainly be very unattractive to do so at the expense of a minimized diffusion length of charge carriers since this would strongly decrease the photovoltaic performance of the materials. This would at least have to be mentioned in the text.

Reviewer #1

1. Starting with the presentation, I do think that the structure of the manuscript would benefit from some reshaping; currently I do not find the presentation accessible to a general readership as Nature Communications publications require. In particular I think the description of the fitting models used for rationalizing the phase-segregation process are not very clear and are quite difficult to follow.

To address this reviewer concern, we have made substantial edits to the manuscript to improve its readability. This has entailed a sizable restructuring of the text as well as numerous editorial changes to render the manuscript more accessible to the lay reader. Additional changes include revising **Figure 2** to improve its clarity so that it better conveys the physical processes described by our kinetic model. Accompanying this response are therefore two versions of the manuscript. The first is the revised manuscript without highlights, the second is a highlighted version, which shows all of the (significant) text changes that have been made. We hope that the reviewer will agree that the clarity of the text has been greatly improved.

2. a) The authors specify that the formation energies of the mixed halide perovskites are calculated using DFT (Figure S3). How are the mixed phases constructed for each concentration? From the computational details authors mention that 'supercells' of 9.3x12.7x8.6 angstrom were used in calculations. However, this is in fact not a supercell, but actually (roughly) the size of the unit cell of MAPbI₃ in the orthorhombic phase (as measured by Baikie et al, *J Mater Chem A* 2013, 1, 5628-5641). It is not clear from the text if the authors use this structure to simulate the mixed compounds, but if so, this approach would bring about significant errors, because they would in fact impose through periodic boundary conditions an ordered arrangement of Br/I configurations which is not evidenced experimentally. The only two ways to ensure such errors do not occur are either to consider (real) supercells of increasing sizes and study multiple (symmetry inequivalent) configurations, or to employ the virtual crystal approximation.

To address this concern, we have completely rewritten **Supplementary Note 4** (not included here for brevity) to fully describe our computational protocol and to justify the approximations made.

Both computational expediency and the results themselves lead us to consider cells containing four MAPbX₃ formula units (FUs), as this cell is of sufficient size to capture the first-nearest-neighbor interactions expected to dominate mixing energies. Our contention [as well as that of Brivio et al. (ref. 9 of the main text); *Thermodynamic origin of photoinstability in the CH₃NH₃Pb(I_{1-x}Br_x)₃ hybrid halide perovskite alloy*. *J. Phys. Chem. Lett.* 2016, 7, 1083-1087] is that the experimentally-observed disordered arrangements are precisely due to very small mixing energies, so that configurational entropy favors disorder. We compute mixing energies relative to the pure phases on the order of < 0.01 eV/FU. Brivio et al. report similar values despite choosing different representations of the mixed halide structures.

As the reviewer suggests, it is possible that there may be some significantly lower energy ordered configuration that we and Brivio et al. have either missed or that cannot be captured within the computational cells. We of course cannot rule this out, and larger supercells alone would not answer the question, as one would have to identify supercells commensurate with the stable structure. However, if this conjecture was true, one would expect to see evidence of it in the structures we have considered, that is, that we would find structures with substantial negative mixing energies that point to the existence of other, even lower energy structures. In fact, Brivio et al. do report a stable ordered structure at $x = \frac{2}{3}$, but even here the mixing energy is quite small relative to the 0.2 eV band gap difference between the iodide and mixed phase. It is this difference that matters for our model. The only practical consequence of the exact mixing energies is on the estimate of n_{max} .

In developing our response to this comment, we discovered that the original manuscript contained an inconsistency between the formation energy and entropy normalizations. In addition to rewriting **Supplementary Note 4**, we have therefore corrected **Supplementary Figure S5**. The changes made decrease the estimated value of n_{max} . We have updated the main text to reflect this correction as well.

3. *b) Another problem in the electronic structure calculations of MAPI is the orientation of the MA cations - how do they decide on the orientation of the MA cation in both the separate calculations of MAPI and MAPB, and the mixed compound? The choice for orientation of the MA cation can distort significantly the structure of the PbX6 network and can sometimes even induce large changes in the electronic structure. In particular, the band alignment shown in Figure S3, which looks to have a valence band offset of 0.1-0.2 eV is highly questionable without a comprehensive discussion and evidence on how these atomistic models were actually built, and how the choices made actually impact the computational conclusions.*

The orientation of the methylammonium (MA) cations is indeed an additional degree of freedom in the compounds. Full consideration of this degree of freedom would greatly increase the cost of the calculations and is unlikely to alter our conclusions. To compute the formation energies, we imposed a consistent orientation across all compositions. This is an approximation *identical* to that applied by Brivio et. al [*Thermodynamic origin of photoinstability in the $CH_3NH_3Pb(I_{1-x}Br_x)_3$ hybrid halide perovskite alloy*. J. Phys. Chem. Lett. 2016. 7, 1083-1087]. It is possible that the 0 K DFT energies could be sensitive to the choice of orientation. However, at finite temperature, the MA cations are dynamic and we expect that their contribution to the free energy will largely cancel across the composition range. There is no reason to expect MA orientations to alter the essential conclusion that mixed halide formation energies are of the order of the excitation energy difference between the mixed and MAPbI₃ compositions.

With regards to the band alignment calculations, we agree that band alignment results can be sensitive to structural model. However, whether the carriers are electrons or holes, the band gap difference will favor charge migration to the iodide region. Further, inference from experimental photoelectron spectroscopy [Butler et. al, *Band alignment of*

the hybrid halide perovskites CH₃NH₃PbCl₃, CH₃NH₃PbBr₃ and CH₃NH₃PbI₃. Mater. Horiz. 2, 2015] indicates that conduction band energies (dominated by Pb 6p states) are approximately constant across the composition range from iodide to bromide, while the valence band energies (dominated by halide p states) vary with composition.

Further, the DFT band alignment calculations are used only as additional support for these observations and are not used directly in the kinetic model. The calculations do not indicate significant charge transfer at the interface and the resultant band alignment is in agreement with the Schottky limit, in which the valence band offset is the difference between the work functions (electron affinity plus the band gap). Thus, the experiments and this approximate model are in agreement with holes as the primary carriers of energy from the mixed to iodide-rich domains.

We appreciate the Reviewer's remarks and have completely rewritten **Supplementary Note 4** to make these ideas more accessible and transparent to the reader.

4. *c) In the section describing the computational methods the authors specify they have/have not used spin-polarized calculations. Why are the authors even considering spin-polarization? MAPI is known to a strong spin-orbit coupling effect (which requires treatment for non-collinear spins), which does reduce the band gap by 1 eV. The authors should justify their choices and/or correct their calculations in this sense.*

We apologize for any confusion here. We intended to write that spin-orbit coupling (and thus necessarily non-collinear magnetism), was included in the total energy calculations. "Spin polarized" was a mis-statement.

For the band alignment calculations, the GGA is well known to give band gaps that are close to experimental values, likely due to cancellation of relativistic and many body effects [Umari et al., *Relativistic GW calculations on CH₃NH₃PbI₃ and CH₃NH₃SnI₃ perovskites for solar cell applications*. Sci. Rep. 4, 2014]. For this reason, we do not include spin-orbit coupling (or, of course, spin polarization) in the band alignment calculations. We have updated the Computational Details in the main text and Supplementary Information to reflect these points.

Specifically, in the main text (page 18, Methods) we now say:

"Spin-orbit coupling and non-collinear magnetism were included in the total energy calculations."

5. *d) In Supporting note 3: 'We estimate the maximal value of n for which the hole localization in the I-rich phase overcomes the free energy of mixing by Eq.S17' – how are the authors reaching this expression?*

The main idea is to construct an energy balance between the energy gain associated with localizing a hole in an iodide-rich domain and the free energy cost of driving phase separation. We have rewritten **Supplementary Note 4** to clarify this analysis.

6. *e) In Supporting note 3 (and through the text) authors use DFT to understand a light-induced process. There is a fundamental problem here in that DFT is a ground state theory – the type of calculations described by the authors do not at any point include excitations (of light or any kind) in the formalism. The authors make no comment on what the implications of this (quite severe approximation) will have on their results, and indeed their conclusions. Therefore, it is really unclear how the authors believe that DFT calculations can in fact provide any support to the experimental data shown throughout the paper.*

The DFT calculations are used exclusively for computing ground state properties in this study. Formation energies are a ground-state property, and the effects of excitations are incorporated using an experimentally derived band gap correlation from Noh et al. *Chemical management for colorful, efficient, and stable inorganic-organic hybrid nanostructured solar cells*. Nano Lett. 2013, 13, 1764-1769. The band alignment calculations are again a ground-state calculation and are used only to support the notion that holes are the primary energy carriers between phases. We believe that the rewrite of **Supplementary Note 4** now makes these points clear.

Reviewer #2

1. *As the manuscript is currently written, I am not able to fully understand what the authors are trying to say. I think the manuscript can likely be modified and become suitable for publication. Below are a list of issues to address.*

To address reviewer 2's concern (as with Reviewer 1), we have made substantial edits to the manuscript to improve its readability. This has entailed a sizable restructuring of the text as well as numerous editorial changes to render the manuscript more accessible to the lay reader. Additional changes include revising **Figure 2** to improve its clarity so that it better conveys physical processes described by the kinetic model. Accompanying this response are therefore two versions of the manuscript. The first is the revised manuscript without highlights, the second is a highlighted version, which shows all of the (significant) text changes that have been made. We hope that Reviewer 2 will agree that the clarity of the text has been greatly improved.

2. *I (subscript Iod) is not well defined at the point where it is first used.*

To address this omission, we have changed the text on page 5 of the revised manuscript to:

“Using emission data from an illuminated MAPb(I_{0.5}Br_{0.5})₃ thin film, **Figure 1c** plots I_{iodide}/I_{sat} as a function of time; I_{iodide} is the integrated intensity of the 725 nm iodide-rich domain emission and I_{sat} is an empirical saturation intensity.”

3. *I do not think Ginsberg et al would agree with the statement “The polaron model additionally suggests that decreasing defect densities in MAPb(1-xBrx)3 films should suppress*

phase separation. However, McGehee and coworkers have previously observed phase separation in single crystals, which exhibit smaller defect densities than their polycrystalline thin film counterparts.” The polarons in Ginsberg’s model are photogenerated charge carriers, not defects.

We thank the reviewer pointing this out. We have removed this sentence from the manuscript. The text has now been changed from

Before

“Despite successes of the polaron model in microscopically rationalizing both phase segregation as well as resulting domain sizes, not all physical observations, such as spectral differences in absorption/emission response as well as non-linear I_{exc} -dependent emission kinetics, are immediately accounted for. The polaron model additionally suggests that decreasing defect densities in MAPb(I_{1-x}Br_x)₃ films should suppress phase separation. However, McGehee and coworkers have previously observed phase separation in single crystals, which exhibit smaller defect densities than their polycrystalline thin film counterparts.[10] Consequently, a better accounting of phase segregation in MAPb(I_{1-x}Br_x)₃ films is needed with the ultimate goal of revealing conditions that will increase their stability under illumination.”

to (page 7, revised manuscript)

After

“Despite successes of the polaron model in rationalizing phase segregation and self-limited domain size growth, many of the ancillary observations, including spectral asymmetries in absorption/emission response as well as non-linear I_{exc} -dependent kinetics, have not been addressed. Consequently, a more complete accounting of light-induced MAPb(I_{1-x}Br_x)₃ phase segregation is needed to identify conditions that can increase the stability of MAPb(I_{1-x}Br_x)₃ films under illumination. ”

4. *I do not think I understand figure 2. I am not 100% sure what a dark circle is supposed to represent. I have no idea at all what a light circle is supposed to represent. I do not know what the red circles are either.*

To address this comment, we have revised **Figure 2** to simplify it and have added a better description of the drawn objects in the figure caption. The following is the revised figure and figure caption.

Figure 2. Schematic of relevant kinetic processes involved in the (a) low and (b) high excitation intensity light-induced phase separation of $\text{MAPb}(\text{I}_{1-x}\text{Br}_x)_3$. Microscopic rate constants ($k_{i,j}$) associated with the underlying kinetic model have been provided. Dark circles denote photogenerated electron-hole pairs. Empty circles denote electron-hole pairs which have induced phase separation. Filled red and white regions represent phase separated iodide-rich and corresponding bromide-rich domains.

5. Line 238 states: “This resulting value is in excellent agreement with the experimental results of McGehee and Ginsberg.¹⁵” Since McGehee and Ginsberg were not coworkers and published separate papers, both papers need to be referenced to avoid confusion.

We again thank the reviewer for pointing out this omission. We have now cited McGehee and Ginsberg separately in the main text.

6. Most of the explanation provided in the manuscript on why the phase separation occurs was already stated in the seminal paper by Hoke, McGehee et al. What remains to be explained

is why it is thermodynamically favorable for the perovskite to be mixed in the dark, but phase separated in the light. Brivio et al's DFT calculations fail to explain this reversibility. Ginsberg's Arxiv paper provides a plausible explanation, but is extraordinarily difficult to understand and has not been accepted in a pure-review journal yet. I consider supplementary figure S3 to be the most important part of this manuscript. I certainly would put it in the main part of the manuscript. Unfortunately I do not consider the calculations that go into figure S3 to be adequately explained. It would seem to me that the increase in free energy due to the presence of excited charge carriers would depend on the density of excited charge carriers. I am not able to tell what assumptions regarding this density were made. The calculations to be explained better.

We appreciate the positive comments of the reviewer regarding **Figure S3 (Figure S5** in the current version of the SI). To reiterate, the basic ideas behind the model are simple and consistent with those of Brivio et al. (Reference 9 of the main text, *Thermodynamic origin of photoinstability in the $\text{CH}_3\text{NH}_3\text{Pb}(\text{I}_{1-x}\text{Br}_x)_3$ hybrid halide perovskite alloy*. J. Phys. Chem. Lett. 2016, 7, 1083-1087): mixing energies are small and entropy drives the formation of the mixed phase in the dark. Under illumination, the dependence of band gap on composition favors generation of an iodide-rich phase and, for mass balance, a bromide-rich phase. Although mixing energies will be sensitive to the exact compositional and structural assumptions made, as is evident in the comparisons between our and Brivio's results, these differences are not critical to establishing the basic physical picture behind the tendency of mixed halide perovskites to phase separate under optical illumination.

Our key elaboration has been to incorporate the energy effects associated with band gap differences between phases. The effect of this contribution does indeed depend on the amount of mixed phase that is separated. This balance is the motivation of the n_{max} calculation mentioned in the main text. However, because of model uncertainties, it is more appropriate to consider this as an order of magnitude estimate rather than an exact evaluation. We have completely rewritten **Supplementary Note 4**, including updating the Figure and caption, to make all of our model assumptions clear.

7. *I don't think the take-home message of this manuscript should be that one can avoid the light-induced phase segregation by reducing the charge carrier diffusion length. For a solar cell to be efficient, it must operate at a high voltage, which implies that there is substantial quasi-Fermi level splitting, which in turn implies a high carrier density. One cannot have a good solar cell without having a high carrier density. If there is a high carrier density, the phase separation is going to occur. Improving the stability by making an inefficient solar cell is not a path that we probably want to go down.*

We understand the reviewer's concern that, within the context of *planar* architecture devices, reducing carrier diffusion lengths is not necessarily the path one wants to take in designing a high efficiency solar cell. However, the primary objective of this study has been to develop a deeper understanding of the phase separation process occurring in mixed halide perovskites. From this, we have uncovered critical physical parameters that

can be deliberately tuned in order to control the phase separation. Diffusion length is one of two parameters that we have identified. Excitation intensity is a second.

The former conclusion may find immediate applicability within the context of *mesoporous* devices where diffusion lengths are already small and where power conversion efficiencies are already on the order of 19%. For planar devices, we additionally note in the main text that other material parameters, such the choice of cation, can be varied to help control phase separation since kinetically this affects the reverse phase recovery rate constant. Empirical support for this latter conclusion exists in the literature as discussed in (a) *Cesium lead halide perovskites with improved stability for tandem solar cells*, Beal et al. *J. Phys. Chem. Lett.*, 2016, 7, 746–751 and (b) *A mixed-cation lead mixed-halide perovskite absorber for tandem solar cells*, D. P. McMeekin et al. *Science* 2016, 351, 151-155.

To address the reviewer’s concern, we have therefore added the following text to page 17 of the revised text

“We find that for $\text{MAPb}(\text{I}_{1-x}\text{Br}_x)_3$ thin films under one sun illumination ($I_{\text{exc}}=100 \text{ mW/cm}^2$) $l_{e/h}$ should be smaller than $\sim 13 \text{ nm}$ to suppress phase segregation. This insight could therefore aid the development of mesoporous hybrid perovskite solar cells where carrier diffusion lengths are already small and where power conversion efficiencies are currently $\sim 19\%$.^[24] Additionally, while reducing $l_{e/h}$ in corresponding planar architectures may not be optimal for improving η , the choice of cation (e.g. cesium, formamidinium) also affects k_{reverse} . Hence, this represents an alternative way by which to enhance the photostability of mixed halide perovskite films in planar devices.^[7,25]”

[7] Beal, R. E. *et al.* Cesium lead halide perovskites with improved stability for tandem solar cells. *J. Phys. Chem. Lett.* **7**, 746–751 (2016).

[24] Jeon, N. J. *et al.* Compositional engineering of perovskite materials for high-performance solar cells. *Nature* **517**, 476–480 (2015).

[25] McMeekin, D. P. *et al.* A mixed-cation lead mixed-halide perovskite absorber for tandem solar cells. *Science* **351**, 151–155 (2016).

Reviewer #3

1. (p.3) *The authors claim the presence of an emission signal at 527 nm which is, however, not shown in Figure 1a (no data below 620 nm) but also is not present in Figure S2a which clearly expands into that range. Further, the authors argue in favour of a dominance of emission at 725 nm over all other signals later in the manuscript. To the opinion of this reviewer such inconsistency should be avoided.*

We thank the reviewer for pointing out this unfortunate omission. To address the stated concern, we have therefore added an inset to **Figure 1a** of the main text, showing the emission at 527 nm which emerges during $\text{MAPb}(\text{I}_{0.5}\text{Br}_{0.5})_3$ illumination. The revised Figure and figure caption are shown below.

Figure 1. (a) Time evolution of MAPb(I_{0.5}Br_{0.5})₃ emission spectra under 405 nm continuous wave excitation ($I_{exc}=20$ mW/cm²). Inset: Emission spectra between 475–600 nm. Stars denote wavelengths of monitored spectral features. (b) Corresponding time evolution of the MAPb(I_{0.5}Br_{0.5})₃ absorption spectra under 405 nm CW excitation ($I_{exc}=25$ mW/cm²). The star denotes the wavelength of the monitored spectral region. Inset: Absorption-based phase separation kinetics from absorption changes at 720 nm where the dashed red line represents an exponential fit to the data. (c) I_{iodide}/I_{sat} under different I_{exc} . Dashed lines are fits using **Equation 3**. The bottom solid red line shows I_{iodide}/I_{sat} when $I_{exc}=40$ mW/cm². (d) I_{exc} -dependent emission-based first order rate constant for phase separation. The dashed red line represents a fit to the data using **Equation 4**.

Additionally, we have added to the Supplementary Information a new figure (**Figure S1**) which shows that the 527 nm and 725 nm emission features exhibit identical growth kinetics under illumination. **Figure S1** and its corresponding caption are reproduced below

Supplementary Figure S1. (a) MAPb(I_{0.5}Br_{0.5})₃ emission spectra after 1 minute of illumination ($I_{exc}=60$ mW/cm², $I_{exc}=405$ nm). (b) Emission kinetics of iodide-rich and bromide-rich domains during illumination.

We have also added to page 2 of the revised manuscript the following text:

“The emission growth kinetics at 527 nm and 725 nm are identical, as shown in the Supplementary Information (SI) **Figure S1**.”

2. (eq.1) Formation of pure MAPbI₃ is claimed whereas ref 10 reported formation of MAPbI_{2.4}Br_{0.6} for all compositions with less than I_{2.4}. Indeed the formation of such composition would even be suggested by the authors’ report of (almost) stable emission spectra for I_{2.64} and I_{2.34} in Figure S1 b and c. Further, disappearance of a well-defined band edge (Figure 1b) without establishment of any other edge characteristic for newly formed perovskite phases would speak in favour of defect formation disturbing the well-defined perovskite band structure and leading to a distribution of superimposed apparent band edges.

We apologize for confusion here. We do not claim separation into pure phases since the chemical composition of the separated phases cannot currently be verified experimentally. Rather, we propose separation into iodide-enriched domains and, as required by mass balance, bromide-enriched domains. While we attempted to be consistent in our use of terminology throughout the manuscript, we believe that confusion arose from inclusion of the *original* Equation 1 in the main text, i.e.

We have now removed this equation from the manuscript to avoid misleading the reader. In addition, we have carefully proofed the main text to ensure that our terminology remains consistent throughout.

Note that nothing in our kinetic model is specific to separation into pure phases. However, the estimate of n_{max} from the DFT-computed formation energies does depend on the composition of the separated phases. The available DFT data and free energy models are not sufficient to determine these compositions. Thus, to estimate n_{max} we

assume separation into pure phases. We have completely rewritten **Supplementary Note 4** to make these assumptions and their consequences clear.

Next, we address the reviewer’s suggestion that the disappearance of a well-defined band edge is more consistent with defect formation. While we agree that static absorption measurements make it hard to define a band edge following prolonged illumination of mixed halide perovskite films, definitive data showing the appearance of new *absorption-related* spectral features linked to phase separation, exists in transient differential absorption spectra we have acquired. In particular, we refer the reviewer to one of our prior publications: *Tracking iodide and bromide ion segregation in mixed halide lead perovskites during photoirradiation*, S. Y. Joon et al., ACS Energy Lett., **2016**, 1, 290–296 where **Figure 3** shows transient absorption spectra of a mixed halide perovskite film following continuous 405 nm irradiation to induce phase separation. Of note are **Figures 3c** and **3d** which show induced bleaches at ~530 nm and ~710 nm which (a) match the emission spectra of bromide- and iodide-rich regions of the film and (b) which are distinct from the starting ~625 nm band gap of the mixed halide. To aid the reviewer, we have reproduced **Figure 3** and its caption here.

Figure 3. (A) Schematic illustration of sample excitation in a pump–probe transient spectrometer. (B–D) Time-resolved difference absorption spectra of $\text{CH}_3\text{NH}_3\text{PbBr}_{1.3}\text{I}_{1.7}$ film recorded following 387 nm laser pulse (pump) excitation (B) before 405 nm CW laser irradiation, (C) after subjecting the sample to 1 min CW laser irradiation, and (D) after subjecting the sample to 40 min CW laser irradiation (405 nm CW laser with 1.7 W/cm^2).

Consequently, to address the reviewer’s concern, we have added to pages 4/5 of the revised main text the following paragraph:

“Evidence for phase separation has additionally been seen in recent transient differential absorption measurements.[12] Namely, under external illumination a ground state bleach[20] maximum associated with the absorption edge of MAPb(I_{0.5}Br_{0.5})₃ (625 nm) disappears and is replaced by two new bleach features, one at 530 nm and another at 720 nm. The former (latter) is associated with the absorption edge of bromide-rich (iodide-rich) perovskite films. These transient absorption results notably agree with the emission data in **Figure 1a**.”

3. *The predictive power of the present DFT calculations is highly overestimated. The authors calculate without consideration of spin polarization. For heavy atoms like Pb this will lead to significantly wrong energy values. The small differences of less than 0.3 eV (e.g., Figure S3) discussed by the authors can easily be within the margin of error of apparent energy differences. Further (main point of criticism !) well-known structural relaxations of methyl ammonium lead iodides/bromides at room temperature are not considered in the present work. The transition from the tetragonal to the cubic phase as most stable structure was observed at I2.4. It would be very surprising if it was just coincidence that the authors observe a very similar composition as critical for the changes observed in the optical properties. The DFT calculations, however, would need to include MD simulation in order to be able to consider such structural relaxations at T>0K.*

As we noted above for Reviewer 1, comment 4, the original submission mistakenly referred to spin-polarization when spin-orbit coupling was meant. The Computational Details have been updated to correct this error.

With regards to the overall DFT calculations, we could not agree more that the reliability of the calculations must not be overstated. Both Reviewers 1 and 2 raise similar points, and we believe that the rewrite of **Supplementary Section 4** addresses this reviewer’s concerns. As correctly noted by Reviewer 3, the exact formation free energies will be sensitive to model assumptions, including structural models, lattice relaxations, MA dynamics, halide ordering, interfacial energies, etc..., and, in general, will be exceedingly difficult to predict reliably. However, as originally noted by Brivio et al. (ref. 9 of the main text, *Thermodynamic origin of photoinstability in the CH₃NH₃Pb(I_{1-x}Br_x)₃ hybrid halide perovskite alloy*. J. Phys. Chem. Lett. 2016, 7, 1083-1087) and as supported by the calculations reported here and by experimental observations, formation energies across the composition domain appear to be quite small, approaching zero, irrespective of model assumptions. We combine this observation with the known dependence of the band gap on composition (Noh et. al, *Chemical management for colorful, efficient, and stable inorganic-organic hybrid nanostructured solar cells*. Nano Lett. 2013, 13, 1764-1769) to infer that band gap differences are sufficient to drive separation into different domains. The only quantitative prediction we make from these calculations is for the value of n_{max} , which we acknowledge is an estimate. We hope that these comments along with rewrite of **Supplementary Section 4** have adequately explained our reasoning.

Regarding the issue of structural phase transitions, the reviewer is referring to **Figure S1** of Ref [10] (McGehee et. al) which is reproduced below.

Figure S1. (left) θ – 2θ XRD patterns of $(\text{MA})\text{Pb}(\text{Br}_x\text{I}_{1-x})_3$ thin films showing the 220 (for $x \leq 0.1$) and 200 (for $x \geq 0.2$) diffraction peaks and (right) the pseudo-cubic lattice parameter extracted from the XRD pattern as a function of alloy ratio.

In the diagram, it is shown that I2.4 to I3.0 exists in the tetragonal phase while for values below I2.4 (i.e. I2.4 to I0.0) the hybrid perovskite exists in the cubic phase. In the current study, we have focused on I1.5. We have also shown data for I0.39, I2.34 and I2.64. Consequently, based on the McGehee data we have the following initial crystallographic structures

- I0.39 = cubic
- I1.5 = cubic
- I2.34 = cubic
- I2.64 = tetragonal

Next, the reviewer appears to suggest that under illumination there is a structural phase transition (i.e. from tetragonal to cubic) which induces an apparent redshifting of the emission. Thus the optical response could be due to a structural phase transition and not due to compositional segregation of iodide and bromide anions. The reviewer highlights the minor (~ 3 meV) redshifting of our sample, I2.64, as evidence of this possibility.

There are several reasons why we have not considered this intriguing hypothesis in the theory.

(1) First, I2.64 begins as tetragonal which is consistent with the fact that pure MAPbI_3 (i.e. I3.0) exists in the tetragonal form [Poglitsch et. al, *Dynamic disorder in methylammoniumtrihalogenoplumbates (II) observed by millimeter-wave spectroscopy*, J. Chem. Phys. 1987, 87, 6373-6378]. Hence, as noted by the reviewer, it is unlikely that an optically-induced tetragonal-to-cubic phase transition will occur. We both agree that there should be little to no shift of the emission in this specimen within the context of the reviewer’s hypothesis. Next, I0.39, I1.5, and I2.34 all begin with the cubic structure. This is consistent with the cubic structure of pure MAPbBr_3 [Poglitsch et. al,

Dynamic disorder in methylammoniumtrihalogenoplumbates (II) observed by millimeter-wave spectroscopy, J. Chem. Phys. 1987, 87, 6373-6378].

At this point, the reviewer suggests that a structural phase transition with I0.39, I1.5 and I2.34 could be the origin of a redshift in the bandgap given that cubic and tetragonal possess different E_g values. However, all three specimens already exist in their lowest energy (cubic) form. Thus, one predicts no bandgap shift at all within the scope of the reviewer's hypothesis. Alternatively, the reviewer could be suggesting that there is a *cubic-to-tetragonal* phase transition for I0.39, I1.5 and I2.34. In this scenario, though, one would predict a *blueshift* of the emission since the tetragonal phase possesses a larger bandgap than cubic (noted by the reviewer as well). Hence, the spectral observations we and others have made are inconsistent with the reviewer's phase transition hypothesis.

(2) Next, during illumination, both emission and transient differential absorption measurements show the emergence of two peaks at different energies (associated wavelengths = 527 nm and 725 nm -see response to Reviewer 3 comment 1 above). Consequently, the data shows the emergence of a material that possesses *two* effective bandgaps. This is again inconsistent with the reviewer's hypothesis of a structural phase transition for the entire mixed halide perovskite being responsible for a net redshift of the emission in a single bandgap system.

(3) Then, we note that the effect is reversible with the material returning to its initial mixed halide perovskite form under dark conditions. Consequently, if the reviewer's hypothesis were correct -that illumination causes a phase transition to the most stable crystal structure of the mixed halide material - one would not expect such reversibility to occur. The final state following illumination would be the most stable structure. The experimental observations are again inconsistent with the reviewer's hypothesis.

(4) Finally, the bandgap difference between the cubic and tetragonal forms of MAPbI₃ is relatively small and is on the order of 20 meV (Foley et. al *Temperature dependent energy levels of methylammonium lead iodide perovskite*, Appl. Phys. Lett. 2015, 106, 243904). This ~20 meV difference [tetragonal 300 K $E_g \sim 1.60$ eV; cubic 328 K $E_g \sim 1.62$] is in dramatic contrast to the ~192 meV spectral shifts seen in our experiments.

4. *While it would be of great technical interest to be able to stabilize mixed halide lead perovskites of a given composition, it would certainly be very unattractive to do so at the expense of a minimized diffusion length of charge carriers since this would strongly decrease the photovoltaic performance of the materials. This would at least have to be mentioned in the text.*

We agree with Reviewer 3 and Reviewer 2 above that reducing carrier diffusion lengths is not necessarily the path one wants to take in designing high efficiency planar architecture solar cells. However, the primary objective of this study has been to develop a deeper understanding of the phase separation process occurring in mixed halide perovskites. From this, we have uncovered critical physical parameters that can be deliberately tuned in order to control the phase separation. Diffusion length is one of two parameters that we have identified. Excitation intensity is a second.

The former conclusion may find immediate applicability within the context of *mesoporous* devices where diffusion lengths are already small and where power conversion efficiencies are already on the order of 19%. For planar devices, we additionally note in the main text that other material parameters, such the choice of cation, can be varied to help control phase separation since kinetically this affects the reverse phase recovery rate constant. Empirical support for this latter conclusion exists in the literature as discussed in (a) *Cesium lead halide perovskites with improved stability for tandem solar cells*, Beal et al. *J. Phys. Chem. Lett.*, 2016, 7, 746–751 and (b) *A mixed-cation lead mixed-halide perovskite absorber for tandem solar cells*, D. P. McMeekin et al. *Science* 2016, 351, 151-155.

To address the concerns of Reviewers 3 and 2, we have therefore added the following text to page 17 of the revised text

“We find that for $\text{MAPb}(\text{I}_{1-x}\text{Br}_x)_3$ thin films under one sun illumination ($I_{\text{exc}}=100$ mW/cm^2) $l_{e/h}$ should be smaller than ~ 13 nm to suppress phase segregation. This insight could therefore aid the development of mesoporous hybrid perovskite solar cells where carrier diffusion lengths are already small and where power conversion efficiencies are currently $\sim 19\%$. [24] Additionally, while reducing $l_{e/h}$ in corresponding planar architectures may not be optimal for improving η , the choice of cation (e.g. cesium, formamidinium) also affects k_{reverse} . Hence, this represents an alternative way by which to enhance the photostability of mixed halide perovskite films in planar devices. [7,25]”

[7] Beal, R. E. *et al.* Cesium lead halide perovskites with improved stability for tandem solar cells. *J. Phys. Chem. Lett.* **7**, 746–751 (2016).

[24] Jeon, N. J. et al. Compositional engineering of perovskite materials for high-performance solar cells. *Nature* **517**, 476–480 (2015).

[25] McMeekin, D. P. *et al.* A mixed-cation lead mixed-halide perovskite absorber for tandem solar cells. *Science* **351**, 151–155 (2016).

Reviewers' comments:

Reviewer #1 (Remarks to the Author):

The authors have addressed my comments fully and have made substantial changes to both the manuscript and the supporting information. My recommendation is that the manuscript be published as is in Nature Communications.

Reviewer #2 (Remarks to the Author):

I still find the manuscript to be difficult to read. As I stated in my earlier comments, the essence of what is new in this manuscript is the free energy equations that explain why phase separation is favored under illumination and mixing is favored in the dark. These equations are not highlighted in the main manuscript. Since Nature Communication is a premier journal, I have no choice but to recommend rejecting the manuscript.

Reviewer #3 (Remarks to the Author):

Reviewer #3

The authors have done a great job in revision of their manuscript. To my opinion the paper can be published. Some of my original concerns do no longer exist, some problems, however, ask for minor revisions.

Original comments 1 and 2 have been widely fixed, just some, partly cosmetic, details have to be taken care of.

Original comment 3 asks for detailed linking of the present results to those reported earlier.

Original comment 4 is fixed.

I left my original comments (for reference) in italics, directly referred to the authors' response (normal text) and wrote my present review in boldface text.

1. (p.3) The authors claim the presence of an emission signal at 527 nm which is, however, not shown in Figure 1a (no data below 620 nm) but also is not present in Figure S2a which clearly expands into that range. Further, the authors argue in favour of a dominance of emission at 725 nm over all other signals later in the manuscript. To the opinion of this reviewer such inconsistency should be avoided.

We thank the reviewer for pointing out this unfortunate omission. To address the stated concern, we have therefore added an inset to Figure 1a of the main text, showing the emission at 527 nm which emerges during MAPb(I0.5Br0.5)3 illumination. The revised Figure and figure caption are shown below.

...

Additionally, we have added to the Supplementary Information a new figure (Figure S1) which shows that the 527 nm and 725 nm emission features exhibit identical growth kinetics under illumination. Figure S1 and its corresponding caption are reproduced below

...

We have also added to page 2 of the revised manuscript the following text:
"The emission growth kinetics at 527 nm and 725 nm are identical, as shown in the Supplementary Information (SI) Figure S1."

The authors have clarified their point substantially and I agree with their view. However, there are still some little inconsistencies present in the argument or, perhaps, just in the presentation: The band shown in Figure S1 shows its peak above 750 nm, at clearly longer wavelength than the spectra in Figure 1a. This spectrum cannot simply be referred to as 725 nm. This difference may be caused by a prolonged illumination at increased intensity which should be explicitly stated. This brings me to the readability of Figure 1. The times in (a) can only be roughly deduced from (c), as can the intensities in (c) be deduced from (d). The authors should at least state this (if valid) or explicitly provide the parameters in the plot or legend.

2. (eq.1) Formation of pure MAPbI3 is claimed whereas ref 10 reported formation of MAPbI2.4Br0.6 for all compositions with less than I2.4. Indeed the formation of such composition would even be suggested by the authors' report of (almost) stable emission spectra for I2.64 and I2.34 in Figure S1 b and c. Further, disappearance of a well-defined band edge (Figure 1b) without establishment of any other edge characteristic for newly formed perovskite phases would speak in favour of defect formation disturbing the well-defined perovskite band structure and leading to a distribution of superimposed apparent band edges.

We apologize for confusion here. We do not claim separation into pure phases since the chemical composition of the separated phases cannot currently be verified experimentally. Rather, we propose separation into iodide-enriched domains and, as required by mass balance, bromide-enriched domains. While we attempted to be consistent in our use of terminology throughout the manuscript, we believe that confusion arose from inclusion of the original Equation 1 in the main text, i.e.

We have now removed this equation from the manuscript to avoid misleading the reader. In addition, we have carefully proofed the main text to ensure that our terminology remains consistent throughout.

Note that nothing in our kinetic model is specific to separation into pure phases. However, the estimate of n_{max} from the DFT-computed formation energies does depend on the composition of the separated phases. The available DFT data and free energy models are not sufficient to determine these compositions. Thus, to estimate n_{max} we assume separation into pure phases. We have completely rewritten Supplementary Note 4 to make these assumptions and their consequences clear.

The authors claim that they tried to omit parts which had indicated the formation of pure phases (I or Br only) and aim at referring to bromide-rich and iodide-rich phases. While this would greatly help to avoid deep misunderstandings, I am not sure that the authors really did what they aimed at. In the new equation 1 we still find pure products only. They either consist of ...Br3 or ...I3 whereas ... $(\text{I}_{1-x-y}\text{Br}_x+\text{y})_3$ and ... $(\text{I}_{1-x+z}\text{Br}_x-z)_3$ are claimed in the text. The desired consistency is not reached yet.

Next, we address the reviewer's suggestion that the disappearance of a well-defined band edge is more consistent with defect formation.

... ..

... .. The former (latter) is associated with the absorption edge of bromide-rich (iodide-rich) perovskite films. These transient absorption results notably agree with the emission data in Figure

1a.”

The authors have argued very clearly and I agree with them and withdraw my original hypothesis of defect formation.

3. The predictive power of the present DFT calculations is highly overestimated. The authors calculate without consideration of spin polarization. For heavy atoms like Pb this will lead to significantly wrong energy values. The small differences of less than 0.3 eV (e.g., Figure S3) discussed by the authors can easily be within the margin of error of apparent energy differences. Further (main point of criticism !) well-known structural relaxations of methyl ammonium lead iodides/bromides at room temperature are not considered in the present work. The transition from the tetragonal to the cubic phase as most stable structure was observed at I2.4. It would be very surprising if it was just coincidence that the authors observe a very similar composition as critical for the changes observed in the optical properties. The DFT calculations, however, would need to include MD simulation in order to be able to consider such structural relaxations at $T > 0\text{K}$. 

As we noted above

... ..

... .. We hope that these comments along with rewrite of Supplementary Section 4 have adequately explained our reasoning.

The authors have revised their method part to a sufficient degree in order to now clearly express their argument, assumptions and limits.

Regarding the issue of structural phase transitions, the reviewer is referring to Figure S1 of Ref [10] (McGehee et. al) which is reproduced below.

... ..

... .. This ~ 20 meV difference [tetragonal 300 K $E_g \sim 1.60$ eV; cubic 328 K $E_g \sim 1.62$] is in dramatic contrast to the ~ 192 meV spectral shifts seen in our experiments.

The authors have clearly demonstrated that a structural transition in any given homogeneous phase can not be the origin of their observations. This, however, was not the center of my main concern. I still think that the authors would do themselves a favor in explicitly mentioning the different stability of structures for different compositions. The fact of a stable I2.64 and I2.34 and the formation of iodide- and bromide-enriched phases for other compositions to my opinion seems to indicate the formation of the iodide-rich phase I2.4 (and accordingly bromide-enriched phases) from compositions poor in iodine in this work as also observed earlier (ref 10). In this context I would not so much refer to their Figure S1 but rather to the text in the second column of page 614 of the article. The authors can not claim to have established a new phenomenon at this point but, rather, provide additional insight into this phenomenon.

4. While it would be of great technical interest to be able to stabilize mixed halide lead perovskites of a given composition, it would certainly be very unattractive to do so at the expense of a minimized diffusion length of charge carriers since this would strongly decrease the photovoltaic performance of the materials. This would at least have to be mentioned in the text.

We agree with Reviewer 3 and Reviewer 2 above that reducing carrier diffusion lengths is not necessarily the path one wants to take in designing high efficiency planar architecture solar cells.

... ..

... ..

... .. Hence, this represents an alternative way by which to enhance the photostability of mixed halide perovskite films in planar devices.[7,25]”

The authors have revised this part to a sufficient degree in order to now clearly express their arguments and conclusions.

Reviewer #1

A. Original comments and response

1. *Starting with the presentation, I do think that the structure of the manuscript would benefit from some reshaping; currently I do not find the presentation accessible to a general readership as Nature Communications publications require. In particular I think the description of the fitting models used for rationalizing the phase-segregation process are not very clear and are quite difficult to follow.*

To address this reviewer concern, we have made substantial edits to the manuscript to improve its readability. This has entailed a sizable restructuring of the text as well as numerous editorial changes to render the manuscript more accessible to the lay reader. Additional changes include revising **Figure 2** to improve its clarity so that it better conveys the physical processes described by our kinetic model. Accompanying this response are therefore two versions of the manuscript. The first is the revised manuscript without highlights, the second is a highlighted version, which shows all of the (significant) text changes that have been made. We hope that the reviewer will agree that the clarity of the text has been greatly improved.

2. *a) The authors specify that the formation energies of the mixed halide perovskites are calculated using DFT (Figure S3). How are the mixed phases constructed for each concentration? From the computational details authors mention that ‘supercells’ of 9.3x12.7x8.6 angstrom were used in calculations. However, this is in fact not a supercell, but actually (roughly) the size of the unit cell of MAPI in the orthorhombic phase (as measured by Baikie et al, J Mater Chem A 2013, 1, 5628-5641). It is not clear from the text if the authors use this structure to simulate the mixed compounds, but if so, this approach would bring about significant errors, because they would in fact impose through periodic boundary conditions an ordered arrangement of Br/I configurations which is not evidenced experimentally. The only two ways to ensure such errors do not occur are either to consider (real) supercells of increasing sizes and study multiple (symmetry inequivalent) configurations, or to employ the virtual crystal approximation.*

To address this concern, we have completely rewritten **Supplementary Note 4** (not included here for brevity) to fully describe our computational protocol and to justify the approximations made.

Both computational expediency and the results themselves lead us to consider cells containing four MAPbX₃ formula units (FUs), as this cell is of sufficient size to capture the first-nearest-neighbor interactions expected to dominate mixing energies. Our contention [as well as that of Brivio et al. (ref. 9 of the main text); *Thermodynamic origin of photoinstability in the CH₃NH₃Pb(I_{1-x}Br_x)₃ hybrid halide perovskite alloy*. J. Phys. Chem. Lett. 2016, 7, 1083-1087] is that the experimentally-observed disordered arrangements are precisely due to very small mixing energies, so that configurational entropy favors disorder. We compute mixing energies relative to the pure phases on the order of < 0.01

eV/FU. Brivio et al. report similar values despite choosing different representations of the mixed halide structures.

As the reviewer suggests, it is possible that there may be some significantly lower energy ordered configuration that we and Brivio et al. have either missed or that cannot be captured within the computational cells. We of course cannot rule this out, and larger supercells alone would not answer the question, as one would have to identify supercells commensurate with the stable structure. However, if this conjecture was true, one would expect to see evidence of it in the structures we have considered, that is, that we would find structures with substantial negative mixing energies that point to the existence of other, even lower energy structures. In fact, Brivio et al. do report a stable ordered structure at $x = \frac{2}{3}$, but even here the mixing energy is quite small relative to the 0.2 eV band gap difference between the iodide and mixed phase. It is this difference that matters for our model. The only practical consequence of the exact mixing energies is on the estimate of n_{max} .

In developing our response to this comment, we discovered that the original manuscript contained an inconsistency between the formation energy and entropy normalizations. In addition to rewriting **Supplementary Note 4**, we have therefore corrected **Supplementary Figure S5**. The changes made decrease the estimated value of n_{max} . We have updated the main text to reflect this correction as well.

3. *b) Another problem in the electronic structure calculations of MAPI is the orientation of the MA cations - how do they decide on the orientation of the MA cation in both the separate calculations of MAPI and MAPB, and the mixed compound? The choice for orientation of the MA cation can distort significantly the structure of the PbX6 network and can sometimes even induce large changes in the electronic structure. In particular, the band alignment shown in Figure S3, which looks to have a valence band offset of 0.1-0.2 eV is highly questionable without a comprehensive discussion and evidence on how these atomistic models were actually built, and how the choices made actually impact the computational conclusions.*

The orientation of the methylammonium (MA) cations is indeed an additional degree of freedom in the compounds. Full consideration of this degree of freedom would greatly increase the cost of the calculations and is unlikely to alter our conclusions. To compute the formation energies, we imposed a consistent orientation across all compositions. This is an approximation *identical* to that applied by Brivio et. al [*Thermodynamic origin of photoinstability in the $CH_3NH_3Pb(I_{1-x}Br_x)_3$ hybrid halide perovskite alloy*. J. Phys. Chem. Lett. 2016. 7, 1083-1087]. It is possible that the 0 K DFT energies could be sensitive to the choice of orientation. However, at finite temperature, the MA cations are dynamic and we expect that their contribution to the free energy will largely cancel across the composition range. There is no reason to expect MA orientations to alter the essential conclusion that mixed halide formation energies are of the order of the excitation energy difference between the mixed and MAPbI₃ compositions.

With regards to the band alignment calculations, we agree that band alignment results can be sensitive to structural model. However, whether the carriers are electrons or

holes, the band gap difference will favor charge migration to the iodide region. Further, inference from experimental photoelectron spectroscopy [Butler et. al, *Band alignment of the hybrid halide perovskites CH₃NH₃PbCl₃, CH₃NH₃PbBr₃ and CH₃NH₃PbI₃*. Mater. Horiz. 2, 2015] indicates that conduction band energies (dominated by Pb 6p states) are approximately constant across the composition range from iodide to bromide, while the valence band energies (dominated by halide p states) vary with composition.

Further, the DFT band alignment calculations are used only as additional support for these observations and are not used directly in the kinetic model. The calculations do not indicate significant charge transfer at the interface and the resultant band alignment is in agreement with the Schottky limit, in which the valence band offset is the difference between the work functions (electron affinity plus the band gap). Thus, the experiments and this approximate model are in agreement with holes as the primary carriers of energy from the mixed to iodide-rich domains.

We appreciate the Reviewer's remarks and have completely rewritten **Supplementary Note 4** to make these ideas more accessible and transparent to the reader.

4. c) *In the section describing the computational methods the authors specify they have/have not used spin-polarized calculations. Why are the authors even considering spin-polarization? MAPI is known to a strong spin-orbit coupling effect (which requires treatment for non-collinear spins), which does reduce the band gap by 1 eV. The authors should justify their choices and/or correct their calculations in this sense.*

We apologize for any confusion here. We intended to write that spin-orbit coupling (and thus necessarily non-collinear magnetism), was included in the total energy calculations. "Spin polarized" was a mis-statement.

For the band alignment calculations, the GGA is well known to give band gaps that are close to experimental values, likely due to cancellation of relativistic and many body effects [Umari et al., *Relativistic GW calculations on CH₃NH₃PbI₃ and CH₃NH₃SnI₃ perovskites for solar cell applications*. Sci. Rep. 4, 2014]. For this reason, we do not include spin-orbit coupling (or, of course, spin polarization) in the band alignment calculations. We have updated the Computational Details in the main text and Supplementary Information to reflect these points.

Specifically, in the main text (page 18, Methods) we now say:

"Spin-orbit coupling and non-collinear magnetism were included in the total energy calculations."

5. d) *In Supporting note 3: 'We estimate the maximal value of n for which the hole localization in the I-rich phase overcomes the free energy of mixing by Eq.S17' – how are the authors reaching this expression?*

The main idea is to construct an energy balance between the energy gain associated with localizing a hole in an iodide-rich domain and the free energy cost of driving phase separation. We have rewritten **Supplementary Note 4** to clarify this analysis.

6. *e) In Supporting note 3 (and through the text) authors use DFT to understand a light-induced process. There is a fundamental problem here in that DFT is a ground state theory – the type of calculations described by the authors do not at any point include excitations (of light or any kind) in the formalism. The authors make no comment on what the implications of this (quite severe approximation) will have on their results, and indeed their conclusions. Therefore, it is really unclear how the authors believe that DFT calculations can in fact provide any support to the experimental data shown throughout the paper.*

The DFT calculations are used exclusively for computing ground state properties in this study. Formation energies are a ground-state property, and the effects of excitations are incorporated using an experimentally derived band gap correlation from Noh et al. *Chemical management for colorful, efficient, and stable inorganic-organic hybrid nanostructured solar cells*. Nano Lett. 2013, 13, 1764-1769. The band alignment calculations are again a ground-state calculation and are used only to support the notion that holes are the primary energy carriers between phases. We believe that the rewrite of **Supplementary Note 4** now makes these points clear.

B. After second round of reviews

The authors have addressed my comments fully and have made substantial changes to both the manuscript and the supporting information. My recommendation is that the manuscript be published as is in Nature Communications.

Reviewer #2

A. Original comments and response

1. *As the manuscript is currently written, I am not able to fully understand what the authors are trying to say. I think the manuscript can likely be modified and become suitable for publication. Below are a list of issues to address.*

To address reviewer 2's concern (as with Reviewer 1), we have made substantial edits to the manuscript to improve its readability. This has entailed a sizable restructuring of the text as well as numerous editorial changes to render the manuscript more accessible to the lay reader. Additional changes include revising **Figure 2** to improve its clarity so that it better conveys physical processes described by the kinetic model. Accompanying this response are therefore two versions of the manuscript. The first is the revised manuscript without highlights, the second is a highlighted version, which shows all of the (significant) text changes that have been made. We hope that Reviewer 2 will agree that the clarity of the text has been greatly improved.

2. *I (subscript I_{od}) is not well defined at the point where it is first used.*

To address this omission, we have changed the text on page 5 of the revised manuscript to:

“Using emission data from an illuminated MAPb(I_{0.5}Br_{0.5})₃ thin film, **Figure 1c** plots $I_{\text{iodide}}/I_{\text{sat}}$ as a function of time; I_{iodide} is the integrated intensity of the 725 nm iodide-rich domain emission and I_{sat} is an empirical saturation intensity.”

3. *I do not think Ginsberg et al would agree with the statement “The polaron model additionally suggests that decreasing defect densities in MAPb(I_{1-x}Br_x)₃ films should suppress phase separation. However, McGehee and coworkers have previously observed phase separation in single crystals, which exhibit smaller defect densities than their polycrystalline thin film counterparts.” The polarons in Ginsberg's model are photogenerated charge carriers, not defects.*

We thank the reviewer pointing this out. We have removed this sentence from the manuscript. The text has now been changed from

Before

“Despite successes of the polaron model in microscopically rationalizing both phase segregation as well as resulting domain sizes, not all physical observations, such as spectral differences in absorption/emission response as well as non-linear I_{exc} -dependent emission kinetics, are immediately accounted for. The polaron model additionally suggests that decreasing defect densities in MAPb(I_{1-x}Br_x)₃ films should suppress phase separation. However, McGehee and coworkers have previously observed phase separation in single crystals, which exhibit smaller defect densities than their polycrystalline thin film counterparts.[10] Consequently, a better accounting of phase

segregation in $\text{MAPb}(\text{I}_{1-x}\text{Br}_x)_3$ films is needed with the ultimate goal of revealing conditions that will increase their stability under illumination.”

to (page 7, revised manuscript)

After

“Despite successes of the polaron model in rationalizing phase segregation and self-limited domain size growth, many of the ancillary observations, including spectral asymmetries in absorption/emission response as well as non-linear I_{exc} -dependent kinetics, have not been addressed. Consequently, a more complete accounting of light-induced $\text{MAPb}(\text{I}_{1-x}\text{Br}_x)_3$ phase segregation is needed to identify conditions that can increase the stability of $\text{MAPb}(\text{I}_{1-x}\text{Br}_x)_3$ films under illumination. ”

4. *I do not think I understand figure 2. I am not 100% sure what a dark circle is supposed to represent. I have no idea at all what a light circle is supposed to represent. I do not know what the red circles are either.*

To address this comment, we have revised **Figure 2** to simplify it and have added a better description of the drawn objects in the figure caption. The following is the revised figure and figure caption.

Figure 2. Schematic of relevant kinetic processes involved in the (a) low and (b) high excitation intensity light-induced phase separation of MAPb(I_{1-x}Br_x)₃. Microscopic rate constants ($k_{i,j}$) associated with the underlying kinetic model have been provided. Dark circles denote photogenerated electron-hole pairs. Empty circles denote electron-hole pairs which have induced phase separation. Filled red and white regions represent phase separated iodide-rich and corresponding bromide-rich domains.

5. Line 238 states: “This resulting value is in excellent agreement with the experimental results of McGehee and Ginsberg.¹⁵” Since McGehee and Ginsberg were not coworkers and published separate papers, both papers need to be referenced to avoid confusion.

We again thank the reviewer for pointing out this omission. We have now cited McGehee and Ginsberg separately in the main text.

6. Most of the explanation provided in the manuscript on why the phase separation occurs was already stated in the seminal paper by Hoke, McGehee et al. What remains to be explained

is why it is thermodynamically favorable for the perovskite to be mixed in the dark, but phase separated in the light. Brivio et al's DFT calculations fail to explain this reversibility. Ginsberg's Arxiv paper provides a plausible explanation, but is extraordinarily difficult to understand and has not been accepted in a pure-review journal yet. I consider supplementary figure S3 to be the most important part of this manuscript. I certainly would put it in the main part of the manuscript. Unfortunately I do not consider the calculations that go into figure S3 to be adequately explained. It would seem to me that the increase in free energy due to the presence of excited charge carriers would depend on the density of excited charge carriers. I am not able to tell what assumptions regarding this density were made. The calculations to be explained better.

We appreciate the positive comments of the reviewer regarding **Figure S3 (Figure S5** in the current version of the SI). To reiterate, the basic ideas behind the model are simple and consistent with those of Brivio et al. (Reference 9 of the main text, *Thermodynamic origin of photoinstability in the $\text{CH}_3\text{NH}_3\text{Pb}(\text{I}_{1-x}\text{Br}_x)$ hybrid halide perovskite alloy*. J. Phys. Chem. Lett. 2016, 7, 1083-1087): mixing energies are small and entropy drives the formation of the mixed phase in the dark. Under illumination, the dependence of band gap on composition favors generation of an iodide-rich phase and, for mass balance, a bromide-rich phase. Although mixing energies will be sensitive to the exact compositional and structural assumptions made, as is evident in the comparisons between our and Brivio's results, these differences are not critical to establishing the basic physical picture behind the tendency of mixed halide perovskites to phase separate under optical illumination.

Our key elaboration has been to incorporate the energy effects associated with band gap differences between phases. The effect of this contribution does indeed depend on the amount of mixed phase that is separated. This balance is the motivation of the n_{max} calculation mentioned in the main text. However, because of model uncertainties, it is more appropriate to consider this as an order of magnitude estimate rather than an exact evaluation. We have completely rewritten **Supplementary Note 4**, including updating the Figure and caption, to make all of our model assumptions clear.

7. *I don't think the take-home message of this manuscript should be that one can avoid the light-induced phase segregation by reducing the charge carrier diffusion length. For a solar cell to be efficient, it must operate at a high voltage, which implies that there is substantial quasi-Fermi level splitting, which in turn implies a high carrier density. One cannot have a good solar cell without having a high carrier density. If there is a high carrier density, the phase separation is going to occur. Improving the stability by making an inefficient solar cell is not a path that we probably want to go down.*

We understand the reviewer's concern that, within the context of *planar* architecture devices, reducing carrier diffusion lengths is not necessarily the path one wants to take in designing a high efficiency solar cell. However, the primary objective of this study has been to develop a deeper understanding of the phase separation process occurring in mixed halide perovskites. From this, we have uncovered critical physical parameters that

can be deliberately tuned in order to control the phase separation. Diffusion length is one of two parameters that we have identified. Excitation intensity is a second.

The former conclusion may find immediate applicability within the context of *mesoporous* devices where diffusion lengths are already small and where power conversion efficiencies are already on the order of 19%. For planar devices, we additionally note in the main text that other material parameters, such the choice of cation, can be varied to help control phase separation since kinetically this affects the reverse phase recovery rate constant. Empirical support for this latter conclusion exists in the literature as discussed in (a) *Cesium lead halide perovskites with improved stability for tandem solar cells*, Beal et al. *J. Phys. Chem. Lett.*, 2016, 7, 746–751 and (b) *A mixed-cation lead mixed-halide perovskite absorber for tandem solar cells*, D. P. McMeekin et al. *Science* 2016, 351, 151-155.

To address the reviewer’s concern, we have therefore added the following text to page 17 of the revised text

“We find that for $\text{MAPb}(\text{I}_{1-x}\text{Br}_x)_3$ thin films under one sun illumination ($I_{\text{exc}}=100 \text{ mW/cm}^2$) $l_{e/h}$ should be smaller than $\sim 13 \text{ nm}$ to suppress phase segregation. This insight could therefore aid the development of mesoporous hybrid perovskite solar cells where carrier diffusion lengths are already small and where power conversion efficiencies are currently $\sim 19\%$.^[24] Additionally, while reducing $l_{e/h}$ in corresponding planar architectures may not be optimal for improving , the choice of cation (e.g. cesium, formamidinium) also affects k_{reverse} . Hence, this represents an alternative way by which to enhance the photostability of mixed halide perovskite films in planar devices.^[7,25]”

[7] Beal, R. E. *et al.* Cesium lead halide perovskites with improved stability for tandem solar cells. *J. Phys. Chem. Lett.* **7**, 746–751 (2016).

[24] Jeon, N. J. et al. Compositional engineering of perovskite materials for high-performance solar cells. *Nature* **517**, 476–480 (2015).

[25] McMeekin, D. P. *et al.* A mixed-cation lead mixed-halide perovskite absorber for tandem solar cells. *Science* **351**, 151–155 (2016).

B. After second round of reviews

I still find the manuscript to be difficult to read. As I stated in my earlier comments, the essence of what is new in this manuscript is the free energy equations that explain why phase separation is favored under illumination and mixing is favored in the dark. These equations are not highlighted in the main manuscript. Since Nature Communication is a premier journal, I have no choice but to recommend rejecting the manuscript.

To address Reviewer 2’s concern we have done the following. First, we have re-edited the entire main text to ensure that it is readable and is pedagogical. Next, we have completely re-written the section on DFT calculations that support the kinetic model, starting on page 10 of the main text. As part of the rewrite we have:

- Re-expressed the original **Equation 1** to make its meaning more apparent to the reader

- Reintroduced the n_{max} expression from the SI back into the main text. This now appears as **Equation 2**. Note that this Equation has also be re-expressed to make its meaning more apparent to the reader.
- Re-introduced the DFT-derived theory figure into the main text. This now appears as **Figure 3**. It can still be found as **Figure S5** of the SI. We also take pains to describe the various predictions shown in **Figure 3** of the main text and as part of this, the captions/symbolic notation used in **Figure 3** have all been improved for clarity.
- Completely re-written **Supplementary Note 4** of the SI in order to improve its pedagogy and clarity.

We hope that this complete re-write of the DFT section, both in the main text and SI (**Supplementary Note 4**), is now satisfactory to the Reviewer and clearly describes how entropy stabilizes the formation of $\text{MAPb}(\text{I}_{1-x}\text{Br}_x)_3$ while bandgap differences between $\text{MAPb}(\text{I}_{1-x}\text{Br}_x)_3$ and MAPbI_3 (or iodide-enriched domains) drive phase separation upon illumination. We appreciate the Reviewer's continued insistence on clarity as we feel that the resulting text has been noticeably improved.

Reviewer #3

A1. Original comments and response

1. (p.3) The authors claim the presence of an emission signal at 527 nm which is, however, not shown in Figure 1a (no data below 620 nm) but also is not present in Figure S2a which clearly expands into that range. Further, the authors argue in favour of a dominance of emission at 725 nm over all other signals later in the manuscript. To the opinion of this reviewer such inconsistency should be avoided.

We thank the reviewer for pointing out this unfortunate omission. To address the stated concern, we have therefore added an inset to **Figure 1a** of the main text, showing the emission at 527 nm which emerges during MAPb(I_{0.5}Br_{0.5})₃ illumination. The revised Figure and figure caption are shown below.

Figure 1. (a) Time evolution of MAPb(I_{0.5}Br_{0.5})₃ emission spectra under 405 nm continuous wave excitation ($I_{exc} = 20 \text{ mW/cm}^2$). Inset: Emission spectra between 475-600 nm. Stars denote wavelengths of monitored spectral features. (b) Corresponding time evolution of the MAPb(I_{0.5}Br_{0.5})₃ absorption spectra under 405 nm CW excitation ($I_{exc} = 25 \text{ mW/cm}^2$). The star denotes the wavelength of the monitored spectral region. Inset: Absorption-based phase separation kinetics from absorption changes at 720 nm where the dashed red line represents an exponential fit to the data. (c) I_{iodide}/I_{sat} under different I_{exc} . Dashed lines are fits using **Equation 3**. The bottom solid red line shows I_{iodide}/I_{sat} when $I_{exc} = 40 \text{ mW/cm}^2$. (d) I_{exc} -dependent emission-based first order rate constant for phase separation. The dashed red line represents a fit to the data using **Equation 4**.

Additionally, we have added to the Supplementary Information a new figure (**Figure S1**) which shows that the 527 nm and 725 nm emission features exhibit identical growth kinetics under illumination. **Figure S1** and its corresponding caption are reproduced below

Supplementary Figure S1. (a) MAPb(I_{0.5}Br_{0.5})₃ emission spectra after 1 minute of illumination ($I_{exc}=60$ mW/cm², $\lambda_{exc}=405$ nm). (b) Emission kinetics of iodide-rich and bromide-rich domains during illumination.

We have also added to page 2 of the revised manuscript the following text:

“The emission growth kinetics at 527 nm and 725 nm are identical, as shown in the Supplementary Information (SI) **Figure S1**.”

A2. After second round of reviews

The authors have clarified their point substantially and I agree with their view. However, there are still some little inconsistencies present in the argument or, perhaps, just in the presentation:

*The band shown in **Figure S1** shows its peak above 750 nm, at clearly longer wavelength than the spectra in **Figure 1a**. This spectrum cannot simply be referred to as 725 nm. This difference may be caused by a prolonged illumination at a higher excitation intensity of the sample used to acquire the data shown in **Figure S1**. This brings me to the readability of **Figure 1**. The times in (a) can only be roughly deduced from (c), as can the intensities in (c) be deduced from (d). The authors should at least state this (if valid) or explicitly provide the parameters in the plot or legend.*

We thank the reviewer for noticing this discrepancy. The reviewer correctly points out that the maximum of the iodide-rich emission peak in **Figure 1a** is ~725 nm whereas it is closer to ~750 nm in **Figure S1**. This difference is indeed due to prolonged illumination at a higher excitation intensity of the sample used to acquire the data shown in **Figure S1**.

Consequently, to prevent any potential misunderstandings we have clarified the main text by referring to the red spectral feature which appears on illumination as “iodide-rich emission” instead of referring to it using the 725 nm moniker. We now say on pages 2 and 3

“...Evident is a decrease in the native MAPb(I_{0.5}Br_{0.5})₃ emission at $\lambda_{mix}=652$ nm accompanied by a corresponding rise of emission features at ~725 and ~527 nm. The former (latter) is associated with MAPbI₃-like (MAPbBr₃-like) photoluminescence. In

what follows, the text therefore refers to the red (green) spectral feature as emission from iodide-rich (bromide-rich) domains. Furthermore, the emission growth kinetics of iodide- and bromide-rich domains are identical, as shown in the Supplementary Information (SI) (**Figure S1**).”

There should be no further instances of “725 nm” emission in the main text or in the SI. Furthermore, in the experimental section of the main text we have added the following text to ensure that the reader knows that we are using integrated emission intensities from fits to experimental spectra in our kinetics:

“All emission spectra were fit to two Gaussians (one for λ_{mix} and the other for λ_{iodide}) to estimate the integrated emission intensity of iodide-rich domains (I_{iodide}). Integrated emission intensities of bromide-rich domains were obtained using both a longpass (425 nm, Chroma) and a shortpass (532 nm, Semrock) filter coupled to an avalanche photodiode (Perkin Elmer, SPCM-AQR-14).”

Next, to specifically address the reviewer’s request to improve the transparency of **Figure 1**, we have (a) added explicit times and excitation intensities to the caption, (b) removed asterisks in panels a and b not central to the discussion, (c) added a x-axis label to the inset in panel b which was absent in the original version, and (d) added two more traces to panel c so that the data presented covers the full range shown in Panel d. The revised **Figure 1** and corresponding caption have been reproduced below

Figure 1. (a) Time evolution of MAPb(I_{0.5}Br_{0.5})₃ emission spectra under 405 nm continuous wave excitation ($I_{exc}=20$ mW/cm²). Times for selected spectra (from red to purple): 0.05, 1.41, 1.64, 1.69, 1.83, 1.93, 2.26, 2.40, 2.58, 2.68, 2.87, 3.10 s. Inset: Emission spectra between 475-600 nm. (b) Corresponding time evolution of the MAPb(I_{0.5}Br_{0.5})₃ absorption spectra under 405 nm CW excitation ($I_{exc}=25$ mW/cm²). Times for selected absorption spectra (from blue to green): 0, 1, 30 minutes. Inset: Absorption-based phase separation kinetics from absorption changes at 720 nm where the dashed red line represents an exponential fit to the data. (c) I_{iodide}/I_{sat} under different I_{exc} . Black dashed lines are fits using Equation 4. The bottom solid red line shows I_{iodide}/I_{sat} when $I_{exc}=40$ μ W/cm². Excitation intensities for selected curves (from red to purple): 0.27, 0.79, 1.05, 1.46, 1.56, 2.27, 3.81, 5.04, 18.56, 57.28 mW/cm². (d) I_{exc} -dependent emission-based first order rate constant for phase separation. The dashed red line represents a fit to the data using Equation 5.

B1. Original comment and response

2. (eq.1) Formation of pure MAPbI₃ is claimed whereas ref 10 reported formation of MAPbI_{2.4}Br_{0.6} for all compositions with less than I_{2.4}. Indeed the formation of such composition would even be suggested by the authors' report of (almost) stable emission spectra for I_{2.64} and I_{2.34} in Figure S1 b and c. Further, disappearance of a well-defined band edge (Figure 1b) without establishment of any other edge characteristic for newly formed perovskite phases would speak in favour of defect formation disturbing the well-defined perovskite band structure and leading to a distribution of superimposed apparent band edges.

We apologize for confusion here. We do not claim separation into pure phases since the chemical composition of the separated phases cannot currently be verified experimentally. Rather, we propose separation into iodide-enriched domains and, as required by mass balance, bromide-enriched domains. While we attempted to be

consistent in our use of terminology throughout the manuscript, we believe that confusion arose from inclusion of the *original* Equation 1 in the main text, i.e.

We have now removed this equation from the manuscript to avoid misleading the reader. In addition, we have carefully proofed the main text to ensure that our terminology remains consistent throughout.

Note that nothing in our kinetic model is specific to separation into pure phases. However, the estimate of n_{max} from the DFT-computed formation energies does depend on the composition of the separated phases. The available DFT data and free energy models are not sufficient to determine these compositions. Thus, to estimate n_{max} we assume separation into pure phases. We have completely rewritten **Supplementary Note 4** to make these assumptions and their consequences clear.

Next, we address the reviewer's suggestion that the disappearance of a well-defined band edge is more consistent with defect formation. While we agree that static absorption measurements make it hard to define a band edge following prolonged illumination of mixed halide perovskite films, definitive data showing the appearance of new *absorption-related* spectral features linked to phase separation, exists in transient differential absorption spectra we have acquired. In particular, we refer the reviewer to one of our prior publications: *Tracking iodide and bromide ion segregation in mixed halide lead perovskites during photoirradiation*, S. Y. Joon et al., ACS Energy Lett., **2016**, 1, 290–296 where **Figure 3** shows transient absorption spectra of a mixed halide perovskite film following continuous 405 nm irradiation to induce phase separation. Of note are **Figures 3c** and **3d** which show induced bleaches at ~530 nm and ~710 nm which (a) match the emission spectra of bromide- and iodide-rich regions of the film and (b) which are distinct from the starting ~625 nm band gap of the mixed halide. To aid the reviewer, we have reproduced **Figure 3** and its caption here.

Figure 3. (A) Schematic illustration of sample excitation in a pump–probe transient spectrometer. (B–D) Time-resolved difference absorption spectra of $\text{CH}_3\text{NH}_3\text{PbBr}_{1.3}\text{I}_{1.7}$ film recorded following 387 nm laser pulse (pump) excitation (B) before 405 nm CW laser irradiation, (C) after subjecting the sample to 1 min CW laser irradiation, and (D) after subjecting the sample to 40 min CW laser irradiation (405 nm CW laser with 1.7 W/cm^2).

Consequently, to address the reviewer’s concern, we have added to pages 4/5 of the revised main text the following paragraph:

“Evidence for phase separation has additionally been seen in recent transient differential absorption measurements.[12] Namely, under external illumination a ground state bleach[20] maximum associated with the absorption edge of $\text{MAPb}(\text{I}_{0.5}\text{Br}_{0.5})_3$ (625 nm) disappears and is replaced by two new bleach features, one at 530 nm and another at 720 nm. The former (latter) is associated with the absorption edge of bromide-rich (iodide-rich) perovskite films. These transient absorption results notably agree with the emission data in **Figure 1a**.”

B2. After second round of reviews

The authors claim that they tried to omit parts which had indicated the formation of pure phases (I or Br only) and aim at referring to bromide-rich and iodide-rich phases. While this would greatly help to avoid deep misunderstandings, I am not sure that the authors really did what they aimed at. In the new equation 1 we still find pure products only. They either consist of ...Br3 or ...I3 whereas ... $(\text{I}1-x-y\text{Br}x+y)_3$ and ... $(\text{I}1-x+z\text{Br}x-z)_3$ are claimed in the text. The desired consistency is not reached yet.

Regarding defect formation

The authors have argued very clearly and I agree with them and withdraw my original hypothesis of defect formation.

This comment appears in line with that of Reviewer 2 and we understand the desire to have improved clarity in the description of the model. In our response to Reviewer 2, we detail the significant changes that we have made to the theory text associated with DFT calculations used to support the kinetic model. Specifically, we have

- Re-expressed the original **Equation 1** in the main text to make its meaning more apparent to the reader
- Reintroduced the n_{max} expression from the SI back into the main text. This now appears as **Equation 2**. Note that this Equation has also be re-expressed to make its meaning more apparent to the reader.
- Re-introduced the DFT-derived theory figure into the main text. This now appears as **Figure 3**. It can still be found as **Figure S5** of the SI. We also take pains to describe the various predictions shown in **Figure 3** of the main text and as part of this, the captions/symbolic notation used in **Figure 3** have been improved for clarity.
- Completely re-written **Supplementary Note 4** of the SI in order to improve its pedagogy and clarity.

Now regarding how to interpret **Equation 1** -rather than write what appears to be a chemical reaction leading to the formation of pure iodide and pure bromide (as done previously) we show that our actual intent was to express the free energy difference associated with photoexcited $\text{MAPb}(\text{I}_{1-x}\text{Br}_x)_3$ and $\text{MAPbI}_3/\text{MAPbBr}_3$. Although this estimate provides a limit and the actual free energy difference will be slightly smaller in the more probably scenario of iodide- and bromide-enriched domains, the *positive* free energy difference found in all cases suggests that band gap differences between $\text{MAPb}(\text{I}_{1-x}\text{Br}_x)_3$ and MAPbI_3 ultimately favor demixing by overcoming the original entropic driving force resulting in formation of $\text{MAPb}(\text{I}_{1-x}\text{Br}_x)_3$. We hope that this concept has now been brought out more clearly both in the main text and in the SI.

C1. Original comment and response

3. *The predictive power of the present DFT calculations is highly overestimated. The authors calculate without consideration of spin polarization. For heavy atoms like Pb this will lead to significantly wrong energy values. The small differences of less than 0.3 eV (e.g., Figure S3) discussed by the authors can easily be within the margin of error of apparent energy differences. Further (main point of criticism !) well-known structural relaxations of methyl ammonium lead iodides/bromides at room temperature are not considered in the present work. The transition from the tetragonal to the cubic phase as most stable structure was observed at I2.4. It would be very surprising if it was just coincidence that the authors observe a very similar composition as critical for the changes observed in the optical properties. The DFT calculations, however, would need to include MD simulation in order to be able to consider such structural relaxations at $T > 0\text{K}$.*

As we noted above for Reviewer 1, comment 4, the original submission mistakenly referred to spin-polarization when spin-orbit coupling was meant. The Computational Details have been updated to correct this error.

With regards to the overall DFT calculations, we could not agree more that the reliability of the calculations must not be overstated. Both Reviewers 1 and 2 raise similar points, and we believe that the rewrite of **Supplementary Section 4** addresses this reviewer’s concerns. As correctly noted by Reviewer 3, the exact formation free energies will be sensitive to model assumptions, including structural models, lattice relaxations, MA dynamics, halide ordering, interfacial energies, etc..., and, in general, will be exceedingly difficult to predict reliably. However, as originally noted by Brivio et al. (ref. 9 of the main text, *Thermodynamic origin of photoinstability in the $CH_3NH_3Pb(I_{1-x}Br_x)_3$ hybrid halide perovskite alloy*. J. Phys. Chem. Lett. 2016, 7, 1083-1087) and as supported by the calculations reported here and by experimental observations, formation energies across the composition domain appear to be quite small, approaching zero, irrespective of model assumptions. We combine this observation with the known dependence of the band gap on composition (Noh et. al, *Chemical management for colorful, efficient, and stable inorganic-organic hybrid nanostructured solar cells*. Nano Lett. 2013, 13, 1764-1769) to infer that band gap differences are sufficient to drive separation into different domains. The only quantitative prediction we make from these calculations is for the value of n_{max} , which we acknowledge is an estimate. We hope that these comments along with rewrite of **Supplementary Section 4** have adequately explained our reasoning.

Regarding the issue of structural phase transitions, the reviewer is referring to **Figure S1** of Ref [10] (McGehee et. al) which is reproduced below.

Figure S1. (left) θ - 2θ XRD patterns of $(MA)Pb(Br_xI_{1-x})_3$ thin films showing the 220 (for $x \leq 0.1$) and 200 (for $x \geq 0.2$) diffraction peaks and (right) the pseudo-cubic lattice parameter extracted from the XRD pattern as a function of alloy ratio.

In the diagram, it is shown that I2.4 to I3.0 exists in the tetragonal phase while for values below I2.4 (i.e. I2.4 to I0.0) the hybrid perovskite exists in the cubic phase. In the current study, we have focused on I1.5. We have also shown data for I0.39, I2.34 and

I2.64. Consequently, based on the McGehee data we have the following initial crystallographic structures

- I0.39 = cubic
- I1.5 = cubic
- I2.34 = cubic
- I2.64 = tetragonal

Next, the reviewer appears to suggest that under illumination there is a structural phase transition (i.e. from tetragonal to cubic) which induces an apparent redshifting of the emission. Thus the optical response could be due to a structural phase transition and not due to compositional segregation of iodide and bromide anions. The reviewer highlights the minor (~ 3 meV) redshifting of our sample, I2.64, as evidence of this possibility.

There are several reasons why we have not considered this intriguing hypothesis in the theory.

(1) First, I2.64 begins as tetragonal which is consistent with the fact that pure MAPbI₃ (i.e. I3.0) exists in the tetragonal form [Poglitsch et. al, *Dynamic disorder in methylammoniumtrihalogenoplumbates (II) observed by millimeter-wave spectroscopy*, J. Chem. Phys. 1987, 87, 6373-6378]. Hence, as noted by the reviewer, it is unlikely that an optically-induced tetragonal-to-cubic phase transition will occur. We both agree that there should be little to no shift of the emission in this specimen within the context of the reviewer's hypothesis. Next, I0.39, I1.5, and I2.34 all begin with the cubic structure. This is consistent with the cubic structure of pure MAPbBr₃ [Poglitsch et. al, *Dynamic disorder in methylammoniumtrihalogenoplumbates (II) observed by millimeter-wave spectroscopy*, J. Chem. Phys. 1987, 87, 6373-6378].

At this point, the reviewer suggests that a structural phase transition with I0.39, I1.5 and I2.34 could be the origin of a redshift in the bandgap given that cubic and tetragonal possess different E_g values. However, all three specimens already exist in their lowest energy (cubic) form. Thus, one predicts no bandgap shift at all within the scope of the reviewer's hypothesis. Alternatively, the reviewer could be suggesting that there is a *cubic-to-tetragonal* phase transition for I0.39, I1.5 and I2.34. In this scenario, though, one would predict a *blueshift* of the emission since the tetragonal phase possesses a larger bandgap than cubic (noted by the reviewer as well). Hence, the spectral observations we and others have made are inconsistent with the reviewer's phase transition hypothesis.

(2) Next, during illumination, both emission and transient differential absorption measurements show the emergence of two peaks at different energies (associated wavelengths = 527 nm and 725 nm -see response to Reviewer 3 comment 1 above). Consequently, the data shows the emergence of a material that possesses *two* effective bandgaps. This is again inconsistent with the reviewer's hypothesis of a structural phase transition for the entire mixed halide perovskite being responsible for a net redshift of the emission in a single bandgap system.

(3) Then, we note that the effect is reversible with the material returning to its initial mixed halide perovskite form under dark conditions. Consequently, if the reviewer's hypothesis were correct -that illumination causes a phase transition to the most stable crystal structure of the mixed halide material - one would not expect such reversibility to occur. The final state following illumination would be the most stable structure. The experimental observations are again inconsistent with the reviewer's hypothesis.

(4) Finally, the bandgap difference between the cubic and tetragonal forms of MAPbI₃ is relatively small and is on the order of 20 meV (Foley et. al *Temperature dependent energy levels of methylammonium lead iodide perovskite*, Appl. Phys. Lett. 2015, 106, 243904). This ~20 meV difference [tetragonal 300 K E_g ~ 1.60 eV; cubic 328 K E_g ~ 1.62] is in dramatic contrast to the ~192 meV spectral shifts seen in our experiments.

C2. After second round of reviews

Regarding DFT

The authors have revised their method part to a sufficient degree in order to now clearly express their argument, assumptions and limits.

Regarding structural phase transitions

The authors have clearly demonstrated that a structural transition in any given homogeneous phase can not be the origin of their observations. This, however, was not the center of my main concern. I still think that the authors would do themselves a favor in explicitly mentioning the different stability of structures for different compositions. The fact of a stable I2.64 and I2.34 and the formation of iodide- and bromide-enriched phases for other compositions to my opinion seems to indicate the formation of the iodide-rich phase I2.4 (and accordingly bromide-enriched phases) from compositions poor in iodine in this work as also observed earlier (ref 10). In this context I would not so much refer to their Figure S1 but rather to the text in the second column of page 614 of the article. The authors can not claim to have established a new phenomenon at this point but, rather, provide additional insight into this phenomenon.

We now believe that we understand the reviewer's comment. In effect, the reviewer is noting that in the McGehee paper (Reference 10 of the main text) irrespective of what x-value he starts out with (e.g. x=0.1 to x=0.9) he ends up with a material that behaves like x=0.2 (I2.4 in the Reviewer's parlance) in terms of its absorption, emission, and X-ray diffraction. Results of the emission experiment are shown in Figure 2b of the McGehee paper where final (post illumination) emission spectra from x=0.1 to x=0.9 more or less overlap at the same wavelength (a value consistent with that from I2.4). Since *all* of these emission spectra are to the blue of the pure iodide film emission wavelength the reviewer therefore suggests that there is a special stability associated with I2.4.

To address this reviewer point, we have now done the following to the main text to better bring out this point:

- First, we emphasize that we do not know the exact final composition of local iodide-rich phases following illumination and that there could indeed be special stability associated with local compositions consistent with I2.4. The main text on page 7 (bottom) has therefore been changed from

Before

“**Figure 2a** illustrates the microscopic processes that underpin the kinetic model. First, light absorption creates an electron-hole pair. Along **Path 1**, these carriers recombine to produce native MAPb(I_{1-x}Br_x)₃ emission ($\lambda_{mix}=652$ nm, $x=0.5$). Along **Path 2**, photoexcitation provides the driving force to induce local halide anion rearrangement and microscopic MAPb(I_{1-x}Br_x)₃ phase segregation into iodide- and bromide-rich domains.”

to

After

“**Figure 2a** illustrates the microscopic processes that underpin the kinetic model. First, light absorption creates an electron-hole pair. Along **Path 1**, these carriers recombine to produce native MAPb(I_{1-x}Br_x)₃ emission ($\lambda_{mix}=652$ nm, $x=0.5$). Along **Path 2**, photoexcitation provides the driving force to induce local halide anion rearrangement and microscopic MAPb(I_{1-x}Br_x)₃ phase segregation into iodide- and bromide-rich domains. Although the exact final composition of these iodide-rich domains is not known, Hoke *et. al* have suggested existence of a stable iodide-rich phase, following prolonged illumination, with a nominal composition of $x\approx 0.2$.**Error! Bookmark not defined.**”

- Next, in the re-written DFT section (page 11 of revised main text) we now explicitly say

“In this regard, **Figure 3a** shows ΔF^* to be close to zero in the composition region $x<0.1$ with a predominantly iodide-rich composition of MAPb(I_{0.9}Br_{0.1})₃. The estimate is consistent with the results of McGehee *et al.* [10] who have previously found iodide-rich phases, following prolonged photoexcitation, having $x\approx 0.2$ [i.e. MAPb(I_{0.8}Br_{0.2})₃].”

- Finally, we mention this again in the revised SI **Supplementary Note 4** as:

“Taking decomposition into pure phases yields the maximum ΔF_{GS} and thus the free energy cost of phase separation. Separation into iodide- and bromide-enriched phases will yield correspondingly smaller ΔF_{GS} values. **Figure S5a** shows ΔF^* to be near zero in the domain $x<0.1$ with a predominantly iodide-rich phase of composition of MAPb(I_{0.9}Br_{0.1})₃. The estimate is consistent with the results of McGehee *et al.* [11] who have previously suggested that iodide-rich phases following prolonged photoexcitation adopt a composition with $x\approx 0.2$ [i.e. MAPb(I_{0.8}Br_{0.2})₃].”

We hope now that we have addressed the Reviewer’s concern and thank him/her for bringing this point out better in the manuscript.

D1. Original comment and response

4. While it would be of great technical interest to be able to stabilize mixed halide lead perovskites of a given composition, it would certainly be very unattractive to do so at the expense of a minimized diffusion length of charge carriers since this would strongly decrease the photovoltaic performance of the materials. This would at least have to be mentioned in the text.

We agree with Reviewer 3 and Reviewer 2 above that reducing carrier diffusion lengths is not necessarily the path one wants to take in designing high efficiency planar architecture solar cells. However, the primary objective of this study has been to develop a deeper understanding of the phase separation process occurring in mixed halide perovskites. From this, we have uncovered critical physical parameters that can be deliberately tuned in order to control the phase separation. Diffusion length is one of two parameters that we have identified. Excitation intensity is a second.

The former conclusion may find immediate applicability within the context of *mesoporous* devices where diffusion lengths are already small and where power conversion efficiencies are already on the order of 19%. For planar devices, we additionally note in the main text that other material parameters, such the choice of cation, can be varied to help control phase separation since kinetically this affects the reverse phase recovery rate constant. Empirical support for this latter conclusion exists in the literature as discussed in (a) *Cesium lead halide perovskites with improved stability for tandem solar cells*, Beal et al. *J. Phys. Chem. Lett.*, 2016, 7, 746–751 and (b) *A mixed-cation lead mixed-halide perovskite absorber for tandem solar cells*, D. P. McMeekin et al. *Science* 2016, 351, 151-155.

To address the concerns of Reviewers 3 and 2, we have therefore added the following text to page 17 of the revised text

“We find that for $\text{MAPb}(\text{I}_{1-x}\text{Br}_x)_3$ thin films under one sun illumination ($I_{\text{exc}}=100$ mW/cm²) $l_{e/h}$ should be smaller than ~ 13 nm to suppress phase segregation. This insight could therefore aid the development of mesoporous hybrid perovskite solar cells where carrier diffusion lengths are already small and where power conversion efficiencies are currently $\sim 19\%$.^[24] Additionally, while reducing $l_{e/h}$ in corresponding planar architectures may not be optimal for improving η , the choice of cation (e.g. cesium, formamidinium) also affects k_{reverse} . Hence, this represents an alternative way by which to enhance the photostability of mixed halide perovskite films in planar devices.^[7,25]”

[7] Beal, R. E. *et al.* Cesium lead halide perovskites with improved stability for tandem solar cells. *J. Phys. Chem. Lett.* **7**, 746–751 (2016).

[24] Jeon, N. J. et al. Compositional engineering of perovskite materials for high-performance solar cells. *Nature* **517**, 476–480 (2015).

[25] McMeekin, D. P. *et al.* A mixed-cation lead mixed-halide perovskite absorber for tandem solar cells. *Science* **351**, 151–155 (2016).

D2. After second round of reviews

The authors have revised this part to a sufficient degree in order to now clearly express their arguments and conclusions.

Reviewers' comments:

Reviewer #2 (Remarks to the Author):

On line 34, the authors boldly state that they "resolve all known experimental observations regarding the light-induced phase segregation of $\text{MAPb}(\text{I}_{1-x}\text{Br}_x)_3$." Many aspects of the light-induced phase separation are already understood qualitatively. What remains to be done is to provide a precise quantitative explanation. This manuscript attempts to do that. However, in contrast to the author's claim, they do not answer one of the most basic and interesting questions- Why does the iodide-rich phase have 80 % iodide regardless of the iodide content of the original film? Why is that number 80 % and not 70 % or 90 %? If I understand Fig 3a correctly, the authors predict that the iodide-rich phase should have 70 % iodide, which is incorrect. I therefore find that their model fails my most basic validity test.

In Figure 3a, the authors perform thermodynamic calculations for two unit cells. I do not understand why one would choose two unit cells. I would think one would need to know the extent to which the charge carrier was delocalized in order to know how many unit cell should be incorporated. Moreover, when only two unit cells are analyzed, what does it even mean to say that demixing occurred? How can average compositions be defined when only two unit cells are being considered?

The author's model will only work if they can figure out how to define the appropriate length scale over which calculations should be performed. I think that length scale is the size over which the charge carrier might be contained.

Reviewer #3 (Remarks to the Author):

The authors have further improved their manuscript substantially and I do think that the paper can be published. All my primary and secondary comments have been properly dealt with BUT ONE PROBLEM STILL REMAINS that the authors did not tackle and which I consider substantial and fundamental to their work.

In order not to get lost in all the comments, rebuttals etc., let me start anew:

- The authors experimentally confirmed a segregation of mixed halide perovskite phases into bromide- and iodide-rich phases. Fine.
- The authors performed DFT calculations within a given crystal structure and noticed that the smaller bandgap of the iodide-rich phase provides driving force for this phase separation under illumination, but not so in the dark. Relevant finding.
- The authors confirmed segregation up to a certain high content of iodide in the iodide-rich phase, but not further. Fine.
- (and now the problem arises) The authors use their DFT calculations to reason this upper limit of iodide content and DO NOT CONSIDER THE FACT that at just this composition the hybrid perovskite becomes stable in a DIFFERENT CRYSTAL STRUCTURE. Different atomic arrangements, however, can have a substantially bigger influence on overall energy than changes within a given structure. Therefore, this fact has to be considered and the STRUCTURAL RELAXATION CAN NOT BE IGNORED.

It is understood that this relaxation can not be included into the present DFT calculations and that massive changes would have to be included by a complete structural relaxation at each given composition of the perovskite superimposed to the already demanding calculations.

In summary, the authors' calculations provide a valuable contribution to the field and they have the potential of providing an explanation for the observed phenomena. It has to be acknowledged, however, that a huge factor of influence could not be considered but might be relevant, if not dominant in stabilization of the iodide-rich phase ($0.8 \approx x \approx 0.9$). The authors have to include a

statement similar to this one at prominent positions in the paper (e.g., bottom of page 11 and bottom of page 18) in order to not mislead the reader.

Response to Reviewers

Reviewer 3

The authors have further improved their manuscript substantially and I do think that the paper can be published. All my primary and secondary comments have been properly dealt with BUT ONE PROBLEM STILL REMAINS that the authors did not tackle and which I consider substantial and fundamental to their work. In order not to get lost in all the comments, rebuttals etc., let me start anew:

- The authors experimentally confirmed a segregation of mixed halide perovskite phases into bromide- and iodide-rich phases. Fine.

- The authors performed DFT calculations within a given crystal structure and noticed that the smaller bandgap of the iodide-rich phase provides driving force for this phase separation under illumination, but not so in the dark. Relevant finding.

- The authors confirmed segregation up to a certain high content of iodide in the iodide-rich phase, but not further. Fine.

- (and now the problem arises) The authors use their DFT calculations to reason this upper limit of iodide content and DO NOT CONSIDER THE FACT that at just this composition the hybrid perovskite becomes stable in a DIFFERENT CRYSTAL STRUCTURE. Different atomic arrangements, however, can have a substantially bigger influence on overall energy than changes within a given structure. Therefore, this fact has to be considered and the STRUCTURAL RELAXATION CAN NOT BE IGNORED. It is understood that this relaxation can not be included into the present DFT calculations and that massive changes would have to be included by a complete structural relaxation at each given composition of the perovskite superimposed to the already demanding calculations.

In summary, the authors' calculations provide a valuable contribution to the field and they have the potential of providing an explanation for the observed phenomena. It has to be acknowledged, however, that a huge factor of influence could not be considered but might be relevant, if not dominant in stabilization of the iodide-rich phase ($0.8 \approx x \approx 0.9$). The authors have to include a statement similar to this one at prominent positions in the paper (e.g., bottom of page 11 and bottom of page 18) in order to not mislead the reader.

Response to reviewer 3

Reviewer 3 recommends for publication after further acknowledgement that the current model does not account for the potential contributions of "...DIFFERENT CRYSTAL STRUCTURE[s]..." in the ground state phase diagram. This is a fair comment, and we have modified the text on both p. 11 and 18 (highlighted in the revised manuscript) to express the fact that we made one choice of lattice symmetry and to express that a more general survey of symmetries would be necessary to produce a quantitative prediction in the iodide-rich limit.

Specifically, we now say on

Page 11

"...Although the estimate is consistent with the results of McGehee et al.¹⁰ who have previously found iodide-rich phases with $x \approx 0.2$ [i.e. $\text{MAPb}(\text{I}_{0.8}\text{Br}_{0.2})_3$], following prolonged photoexcitation, McGehee et al.¹⁰ as well as Noh et al.⁸ have also suggested that such iodide-enriched phases undergo a crystallographic phase transition from cubic to tetragonal. Consequently, an exact quantitative accounting of the resulting stoichiometry in the iodide-rich limit would require both crystallographic forms to be considered explicitly within the current DFT and thermodynamic modeling wherein such a phase transition could represent a barrier to suppress further iodide phase segregation.^{8,10,}

Page 18

“In summary, we have established quantitative insights into the light-induced phase separation of MAPb(I_{1-x}Br_x)₃ thin films. Bandgap reduction of iodide-rich domains is found to be the driving force that overcomes unfavorable formation energies to induce iodide and bromide segregation. A DFT-based thermodynamic model shows that entropy dominates formation free energies and provides estimates of the initial phase-separated domain size and compositions. These predictions are sensitive to precise structural and entropic models, and quantitative predictions, especially in the iodide-rich limit, would demand a much broader survey of the perovskite crystallographic structure and composition spaces...”

Reviewer 2

On line 34, the authors boldly state that they “resolve all known experimental observations regarding the light-induced phase segregation of MAPb(I_{1-x}Br_x)₃.” Many aspects of the light-induced phase separation are already understood qualitatively. What remains to be done is to provide a precise quantitative explanation. This manuscript attempts to do that. However, in contrast to the author’s claim, they do not answer one of the most basic and interesting questions- Why does the iodide-rich phase have 80 % iodide regardless of the iodide content of the original film? Why is that number 80 % and not 70 % or 90 %? If I understand Fig 3a correctly, the authors predict that the iodide-rich phase should have 70 % iodide, which is incorrect. I therefore find that their model fails my most basic validity test.

In Figure 3a, the authors perform thermodynamic calculations for two unit cells. I do not understand why one would choose two unit cells. I would think one would need to know the extent to which the charge carrier was delocalized in order to know how many unit cell should be incorporated. Moreover, when only two unit cells are analyzed, what does it even mean to say that demixing occurred? How can average compositions be defined when only two unit cells are being considered?

The author’s model will only work if they can figure out how to define the appropriate length scale over which calculations should be performed. I think that length scale is the size over which the charge carrier might be contained.

Response to Reviewer 2

Reviewer 2 notes that the statement that we “resolve all known experimental observations regarding the light-induced phase segregation...” is too strong, and on this point we agree. In the revised text we have moderated this statement as well as any other like statements (highlighted on pages 2 and 7 of the revised manuscript). Specifically, we now say

Page 2

“...In the current study, we rationalize the bulk of experimental observations regarding the light-induced phase segregation of MAPb(I_{1-x}Br_x)₃...”

Page 7

““In what follows, we describe a conceptual framework that rationalizes virtually all current experimental observations regarding MAPb(I_{1-x}Br_x)₃ phase separation...””

Reviewer 2 expresses concerns about the quantitative reliability of the predicted compositions of the phase-separated domains in the iodide-rich limit, a point that is essentially the same as that of Reviewer 3. In no version of the manuscript have we claimed that our DFT-based prediction is quantitatively reliable, and in fact we have gone to great lengths to emphasize the fact that a subtle balance of entropy and energy are at play that are very difficult to predict quantitatively. We have also explicitly included in the main text language (in response to Reviewer 3) to indicate that we have not explicitly considered a tetragonal-to-cubic phase transition at $x \sim 0.2$ which has been suggested by McGehee *et al.* (Reference 10 of the main text) and Noh *et al.* (Reference 8 of the main text) to be a special stability point which limits/suppresses further phase segregation. We therefore believe that the modifications made in response to Reviewer 3, both here and previously, satisfactorily address these concerns given that our primary intent has been to model the *physical*

process driving light-induced phase segregation of mixed halides---not its suggested stoppage in the iodide rich limit below $x \sim 0.2$.

Reviewer 2 also states *“In Figure 3a, the authors perform thermodynamic calculations for two unit cells. I do not understand why one would choose two unit cells. I would think one would need to know the extent to which the charge carrier was delocalized in order to know how many unit cell should be incorporated.”* Later on, they write *“The author’s model will only work if they can figure out how to define the appropriate length scale over which calculations should be performed. I think that length scale is the size over which the charge carrier might be contained.”*

As we describe on p. 11 of the main text and in the Supplementary material, within our model we do not directly calculate formation energies in the presence of charge carriers. Rather, we combine ground-state-computed formation free energies with reliable experimentally observed band edges to estimate formation energies in the presence of an excitation. This formalism is described in the text surrounding Equation 1 and in the equation itself. This approach avoids the need to compute composition-dependent excite-state energies explicitly.

Reviewer 2 further writes *“Moreover, when only two unit cells are analyzed, what does it even mean to say that demixing occurred? How can average compositions be defined when only two unit cells are being considered?”*

“Demixing” is not computed within a single supercell, as such an approach would be computational expensive and, for our purposes, inappropriate. Rather, we compute the ground state energies of the pure and various intermediate compositions within individual supercells of sufficient size to allow us to sample compositions and local orderings likely to contribute to any driving force for demixing. We then construct a formation energy diagram (Figure 3a and unnumbered equation at the beginning of Supplementary section S4) to determine the energy of any given composition relative to the endpoint compositions. Such an approach is widely used in the computational materials community, including in Nature Communications (see e.g. <https://dx.doi.org/10.1038%2Fncomms13814>, Supplementary Figure 12). As we explain in the paragraphs preceding equation S15, our calculations, in line with those of Brivio, find mixing energies to be very small (hence the difficulty in making quantitative predictions), so that entropy drives the mixing of anions at equilibrium. These ideas are fully described in the top paragraph on page 10 and on pp. 6 and 7 in the Supplementary. The consistent use of the 12 anion supercell maximizes numerical error cancellation in between supercells of different composition. It does limit the compositions that can be represented to multiples of $x = 1/12$. Thus, the supercell could not represent a stable phase of some composition incommensurate with factors of $1/12$. However, as we note in the SI, signatures indicating the existence of such a compositional phase are not revealed in our calculations, in those of Brivio et al., or in the experiments themselves.

Reviewers' comments:

Reviewer #2 (Remarks to the Author):

At this point I recommend publishing the manuscript so that the broader community can see the authors' model.

Reviewer #3 (Remarks to the Author):

In this further revised manuscript the authors are willing to admit some limit of their model calculations in explaining the experimental results. I noticed, however, that they still avoid to clearly mention that the formation of a different crystal structure with high iodine content could represent the driving force for the stabilization of an iodine-rich phase and that such explanation could indeed serve as an argument alternative to the authors' argument. According to my understanding the new statements of the authors are developing in the right direction but are still not expressing this fact to a sufficient degree of precision since rather vague statements are made.

Response to Reviewers

Reviewer 3

“In this further revised manuscript the authors are willing to admit some limit of their model calculations in explaining the experimental results. I noticed, however, that they still avoid to clearly mention that the formation of a different crystal structure with high iodine content could represent the driving force for the stabilization of an iodine-rich phase and that such explanation could indeed serve as an argument alternative to the authors’ argument. According to my understanding the new statements of the authors are developing in the right direction but are still not expressing this fact to a sufficient degree of precision since rather vague statements are made.”

Response to reviewer 3

We thank the reviewer for his/her positive comments and now have added the requested specificity to the text on pages 11/12 and 18/19. We now say (new additions highlighted green)

Page 11/12

“...Although the estimate is consistent with the results of McGehee et al.¹⁰ who have previously found iodide-rich phases with $x \approx 0.2$ [i.e. $\text{MAPb}(\text{I}_{0.8}\text{Br}_{0.2})_3$], following prolonged photoexcitation, McGehee et al.¹⁰ as well as Noh et al.⁸ have also suggested that such iodide-enriched phases undergo a crystallographic phase transition from cubic to tetragonal. **It is therefore possible that such a phase transition could stabilize an iodide-rich phase.** Consequently, an exact quantitative accounting **of light-induced phase separation, especially in the iodide-rich limit** would require both crystallographic forms to be considered explicitly within the current DFT and thermodynamic modeling.^{8,10,}”

Page 18/19

“In summary, we have established quantitative insights into the light-induced phase separation of $\text{MAPb}(\text{I}_{1-x}\text{Br}_x)_3$ thin films. Bandgap reduction of iodide-rich domains is found to be the driving force that overcomes unfavorable formation energies to induce iodide and bromide segregation. A DFT-based thermodynamic model shows that entropy dominates formation free energies and provides estimates of the initial phase-separated domain size and compositions. These predictions are sensitive to precise structural and entropic models, and quantitative predictions, especially in the iodide-rich limit, would demand a much broader survey of the perovskite crystallographic structure and composition spaces. **In particular, it is possible that a crystallographic phase transition additionally stabilizes an iodide-rich phase....**”

Reviewers' comments:

Reviewer #3 (Remarks to the Author):

The authors have now successfully dealt with all comments made earlier. I propose publication of their work.